# RDT2: Exploring the Scaling Limit of UMI Data Towards Zero-Shot Cross-Embodiment Generalization

**Songming Liu** [* 1]  **Bangguo Li** [* 1]  **Kai Ma** [* 1]  **Lingxuan Wu** [* 1 2]  **Hengkai Tan** [1]  **Xiao Ouyang** [1]  **Hang Su** [1]
**Jun Zhu** [1]

## Abstract

Vision-Language-Action (VLA) models hold promise for generalist robotics but currently struggle with data scarcity, architectural inefficiencies, and the inability to generalize across different hardware platforms. We introduce RDT2, a robotic foundation model built upon a 7B parameter VLM designed to enable zero-shot deployment on novel embodiments for open-vocabulary tasks. To achieve this, we collected one of the largest open-source robotic datasets—over $10,000$ hours of demonstrations in diverse families—using an enhanced, embodiment-agnostic Universal Manipulation Interface (UMI). Our approach employs a novel three-stage training recipe that aligns discrete linguistic knowledge with continuous control via Residual Vector Quantization (RVQ), flow-matching, and distillation for real-time inference. Consequently, RDT2 becomes one of the first models that simultaneously zero-shot generalizes to unseen objects, scenes, instructions, and even robotic platforms. Besides, it outperforms state-of-the-art baselines in dexterous, long-horizon, and dynamic downstream tasks like playing table tennis.

## 1. Introduction

Vision-Language-Action (VLA) models represent a promising paradigm for achieving generalized embodied intelligence (Team et al., 2024; Kim et al., 2024; Liu et al., 2024; Black et al., 2024; Intelligence et al., 2025). They are particularly well-suited for complex manipulation tasks involving deformable objects and fluids (Ma et al., 2024), which have

long been challenging for traditional control methods due to the difficulty of physical modeling and system identification (Saha & Isto, 2006; Jatavallabhula et al., 2021). However, despite several valuable trials (Ma et al., 2024), current VLA models have not replicated the broad generalization capabilities characteristic of large-scale models in other domains such as Natural Language Processing (NLP) (Achiam et al., 2023; Touvron et al., 2023; Bai et al., 2023; Guo et al., 2025). They often struggle to perform reliably when encountering novel scenes, objects, instructions, or embodiments (Ma et al., 2024), hindering real-world applications.

Developing generalizable VLA models for robotics presents two fundamental challenges. The first is the acquisition of large-scale, diverse datasets. Traditional data collection through teleoperation (Zhao et al., 2023; Fu et al., 2024) is often prohibitively expensive and lacks variety due to the physical constraints and high cost of robotic platforms. In contrast, the Universal Manipulation Interface (UMI) (Chi et al., 2024) provides an embodiment-agnostic, handheld device that enables efficient and low-cost data collection across a multitude of real-world scenarios. The second challenge lies in designing network architectures that can effectively learn from this large-scale robot data. A key difficulty is the inherent multimodality of human-collected demonstrations (Chen et al., 2022; Chi et al., 2023). Prior approaches that model action probabilities via discretization (Brohan et al., 2022; Zitkovich et al., 2023; Kim et al., 2024) are often constrained by the resultant errors and the inefficiency of autoregressive inference. Alternative methods using diffusion models (Chen et al., 2022; Chi et al., 2025; Liu et al., 2024; Black et al., 2024) suffer from slow convergence (Pertsch et al., 2025) and a fundamental mismatch between their continuous probability distributions and the discrete counterparts of knowledge in pre-trained Vision-Language Models (VLMs). Furthermore, a significant tension exists between the growing size of these models and the real-time performance required for robotic tasks. While some distillation techniques have been explored (Chen et al., 2023; Wang et al., 2024b; Prasad et al., 2024), the development of practical methods for large-scale VLA models remains an open problem.

Furthermore, VLA models confront a significant limita-

*Equal contribution [1]Dept. of Comp. Sci & Tech., AI Institute, THU-Bosch Joint Center for ML, Tsinghua University [2]Peng Cheng Laboratory, 518108, China. Correspondence to: <dc-szj@tsinghua.edu.cn>.

*Proceedings of the 43rd International Conference on Machine Learning*, Seoul, South Korea. PMLR 306, 2026. Copyright 2026 by the author(s).

tion for cross-embodiment deployment. Due to variations in physical characteristics across robotic platforms, models trained on one embodiment exhibit poor generalization when transferred to another. While some methods attempt to unify data from different embodiments into a common embedding space (Team et al., 2024; Liu et al., 2024; Wang et al., 2024a; Yang et al., 2024), they still fall short of enabling zero-shot deployment on novel platforms. Consequently, adapting a VLA model to a new robot often necessitates hundreds of hours of data collection and fine-tuning. This substantial cost not only impedes the reproducibility and widespread applicability of VLA research but also curtails the overall progress of the field (Khazatsky et al., 2024; Atreya et al., 2025; Mirchandani et al., 2025).

To address the aforementioned challenges, we introduce *RDT2*, one of the first robotic foundation models for zero-shot deployment on novel embodiments, which can handle open-vocabulary tasks. RDT2 is built upon a 7B pretrained VLM, Qwen2.5-VL (Bai et al., 2025), with specialized action heads and three-stage training strategies for learning from large-scale robotic data. For fast convergence, in Stage 1, we encode the continuous robot actions into discrete tokens with Residual Vector Quantization (RVQ) (Van Den Oord et al., 2017; Esser et al., 2021; Lee et al., 2022) and then train the VLM by minimizing the cross-entropy loss. This also avoids destroying the knowledge stored in the form of discrete probabilities during pre-training. In Stage 2, for expressiveness and efficiency, we employ an action expert to model continuous probability and train it with the flow-matching loss. In Stage 3, we propose a simple yet effective distillation loss and distill the action expert into a single-step generator, achieving ultra-fast inference speed.

Based on the above methods, we were able to train our model, RDT2, on one of the largest open-source UMI datasets, comprising over $10,000$ hours of human demonstrations. This large-scale data collection was made possible by redesigning the UMI hardware with higher-strength materials and high-precision tracking methods to ensure reliability. We fabricated approximately 100 of these enhanced devices and deployed them across more than 100 real-world household environments to capture a diverse range of manipulation tasks. In our experiments, we first evaluated RDT2's zero-shot generalizability across unseen objects, scenes, instructions, and even embodiments. Because UMI provides an embodiment-agnostic physical interface, coupled with large-scale pretraining, RDT2 became one of the first models to achieve combined generalization of the four factors for open-vocabulary tasks. Through experiments with four different model sizes, we discovered that simultaneously scaling up model parameters and data scale yields consistent and predictable performance gains. Besides, our fine-tuning experiments showed that RDT2 outperformed state-of-the-art baselines such as $\pi_0$-FAST (Pertsch et al.,

2025) and $\pi_{0.5}$ (Intelligence et al., 2025) on challenging tasks involving deformable objects, dexterity, long horizons, and high dynamics, such as playing table tennis. Finally, extensive ablation studies demonstrated the effectiveness of the adopted training strategy and design choices.

## 2. Related Work

**Data Pyramid for Robotics.** The landscape of data for robot learning can be conceptualized as a pyramid (Bjorck et al., 2025). At the apex resides teleoperation data, which, gathered via systems like VR (Khazatsky et al., 2024; Cheng et al., 2024; Chen et al., 2025a) or master-slave arms (Zhao et al., 2023; Fu et al., 2024; Aldaco et al., 2024), offers the highest fidelity but is also the most expensive to acquire. Its collection is typically confined to structured laboratory settings (Walke et al., 2023; Fang et al., 2023; Khazatsky et al., 2024; O'Neill et al., 2024; Wu et al., 2024), creating a distributional gap between the training data and real-world applications. Occupying the middle tier is simulation data (Wang et al., 2023; Li et al., 2023; Mu et al., 2024; Chen et al., 2025b); it is inexpensive and scalable but plagued by a significant sim-to-real gap, and the challenge of generating diverse, interactive, and realistic scenarios remains an open problem (Nasiriany et al., 2024; Ren et al., 2024; Zhang et al., 2025). At the base lies the vast repository of internet videos (Ye et al., 2024; Yang et al., 2025; Luo et al., 2025; Feng et al., 2025). Although abundant, this data is unstructured and noisy, and most importantly, lacks the explicit action labels required for the supervised policy training (McCarthy et al., 2025).

**Imitation Learning Models.** Previous models can be broadly classified by their strategy for generalization. A significant body of work focuses on small-scale models (Pari et al., 2021; Florence et al., 2022; Shafiullah et al., 2022; Jang et al., 2022), such as Diffusion Policy (Chen et al., 2022; Chi et al., 2023) and ACT (Zhao et al., 2023), which are typically trained on a per-task basis. While proficient within their specific domains, these models inherently lack the capacity to generalize across diverse tasks or embodiments. To address the cross-embodiment challenge, another line of research leverages the UMI (Chi et al., 2024; Xu et al., 2025) to collect embodiment-agnostic data. However, the limited scale of these datasets constrains the resulting models' performance on open-vocabulary tasks. More recently, the field has seen the emergence of large models like OpenVLA (Kim et al., 2024), RDT-1B (Liu et al., 2024), $\pi_0$ (Black et al., 2024), and $\pi_{0.5}$ (Intelligence et al., 2025). However, since the dataset relies on specific robots, they fall short of embodiment transferability without fine-tuning.

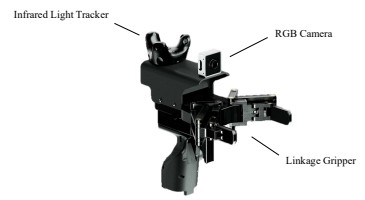

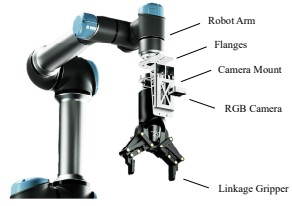

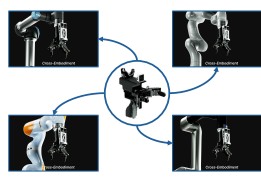

*(a)* Our Re-Designed UMI  *(b)* Easy Deployment  *(c)* Cross-Embodiment Transfer

*Figure 1.* Illustration of our UMI solution. We re-designed the UMI hardware for better consistency and reliability in large-scale data collection. As long as the same model of camera and gripper are installed, the policy trained on the data collected by our UMIs can be zero-shot transferred to various robot arms.

## 3. Problem Formulation and Challenges

We consider the *bimanual manipulation* task in the setting of *language-conditioned imitation learning* for VLA models, which is well-established in the field of robot learning (Stepputtis et al., 2020; Yao et al., 2026). Formally, the task is modeled as a sequential decision-making process. Let $\ell$ denote the a free-form language instruction describing the task. At each time step $t$, an agent is required to take an action chunk $\mathbf{A}_t := (\mathbf{a}_t, \ldots, \mathbf{a}_{t+T_a})$ sampled from $p(\mathbf{A}_t \mid \ell, \mathbf{o}_t)$, where $\mathbf{a}_t \in \mathbb{R}^d$ is the $d$-dimensional action taken at $t$, $T_a$ is the chunk size (Zhao et al., 2023), and $\mathbf{o}_t$ is the RGB observation that the agent accept at $t$. Here, we assume that $\mathbf{o}_t$ already contains all the information needed to make a decision, without considering historical observations $\{\mathbf{o}_i \mid i < t\}$. To obtain a feasible agent, we train a VLA model to learn the distribution $p(\mathbf{A}_t \mid \ell, \mathbf{o}_t)$ from a demonstration dataset of human experts $\mathcal{D} := \{(\ell^{(i)}, \mathbf{o}_t^{(i)}, \mathbf{A}_t^{(i)}) \mid 0 \le t < T^{(i)}, 1 \le i \le N\}$, where $T^{(i)}$ is the $i$-th trajectory length and $N$ is the number of total trajectories.

A given manipulation task is defined by the composition of several key elements: the manipulated *object*, the operational *scene*, the natural language *instruction* from users, and the robotic *embodiment*. Any finite training dataset can only cover a sparse subset of the vast combinatorial space spanned by these elements. A practical VLA model must therefore generalize to unseen compositions of objects, scenes, instructions, and even novel embodiments during deployment. Achieving this compositional generalization, which is crucial for real-world applications, presents two fundamental challenges:

**Challenge of Scaling Up Robotic Data.** It is well-studied in natural language processing and computer vision that increasing dataset scale and diversity improves model generalization (Kaplan et al., 2020; Zhai et al., 2022). However, applying this principle to robotics via teleoperation is currently impractical. The high cost of robotic hardware makes parallel data acquisition, and thus large-scale data collection,

prohibitively expensive. Furthermore, the lack of portability of these systems constrains data acquisition primarily to laboratory or factory settings, severely limiting the diversity and real-world relevance of the collected data. This data scarcity is exacerbated by hardware heterogeneity, as data collected on one robotic platform is often incompatible with others, creating isolated and non-interoperable datasets.

**Challenge of Network Architecture.** According to previous studies (Chen et al., 2022; Chi et al., 2023), human data exhibits significant multimodality, necessitating models that learn a distribution over actions rather than a deterministic mapping. This leads to a choice between discrete and continuous action representations, each with distinct trade-offs. Discrete methods align naturally with the probabilistic outputs of the pretrained VLM, but they suffer from quantization errors and the inefficiency of autoregressive sampling. Conversely, continuous approaches like diffusion models offer more efficient sampling but are hampered by slower training convergence (Pertsch et al., 2025) and risk corrupting the discrete knowledge within the VLM (Deng et al., 2025). A critical challenge, therefore, is to synthesize the advantages of both paradigms. What is more, the real-time performance demanded by robotic tasks makes the efficient deployment of large-scale VLAs a formidable obstacle.

## 4. Hardware and Dataset

To address the data-scaling challenge, we employ UMI (Chi et al., 2024), a portable framework facilitating scalable, in-the-wild data collection. UMI records the 6-DoF end-effector pose and the gripper width using a hand-held device with vision and a tracker. When installing a physically consistent gripper, policies learned from UMI data can be deployed on diverse robotic arms as both vision and structure gaps are minimized across embodiments. Fig. 1 illustrates our UMI solution from design to deployment.

**Re-Designing UMI.** However, the original UMI hardware lacks the reliability requisite for large-scale in-the-wild col-

*Table 1.* Comparison between the original UMI and our redesigned hardware.

| Specification | Naive UMI | Our UMI | Advantage |
|---|---|---|---|
| Fabrication | 3D Printing (PLA / PETG) | CNC (nylon 66 & glass fiber) | Higher stiffness, better machining accuracy and consistency; suitable for long-term, high-frequency data collection |
| Tracking | SLAM | Infrared Light | Better tracking precision for the end-effector 6D pose; more robust to high-speed motion, texture-less backgrounds, and transparent backgrounds |
| End-Effector | Parallel Jaws | Linkage Gripper | More compact structure; improved dexterity and accessibility in tight clearances or clutter |

lection. To address this, we re-engineered the whole system (Fig. 1a) to maximize *structural rigidity*, ensure *drift-free infrared tracking*, and enhance *manipulation dexterity* in cluttered environments. As shown in Tab. 1, these modifications resolve critical pose inconsistencies and limitations of reachability, yielding significantly improved data fidelity; detailed hardware specifications are provided in App. A.

**UMI Dataset at Scale.** Enabled by these hardware advancements, we curate one of the *largest* open-source UMI datasets to date, comprising over $10,000$ hours of manipulation data spanning more than $100$ households. Captured entirely in the wild, our dataset encapsulates a vast distribution of unstructured environments and complex human behaviors, providing a robust substrate for generalist policy learning; we elaborate on the dataset details in App. B.

## 5. Model and Training Pipeline

We introduce RDT2, a VLA model trained through a three-stage pipeline as shown in Fig. 2. In Stage 1 (Sec. 5.1), we discretize the continuous action space into tokens using RVQ and train the VLM backbone via a standard cross-entropy loss. Subsequently, in Stage 2 (Sec. 5.2), we freeze the VLM backbone and train a diffusion-based action expert, leveraging a flow-matching loss to generate continuous actions. This hybrid approach harnesses the benefits of both discretization and diffusion, effectively addressing the challenge of modeling multimodal action distributions. Finally, to resolve the real-time challenge for robotic tasks, the Stage 3 (Sec. 5.3) involves distilling the multi-step action expert into an efficient, single-step generator, thereby enabling rapid inference for our large-scale VLA model. We refer to App. C for hyperparameter and training details.

### 5.1. Stage 1

As previously discussed, diffusion models present two primary issues for VLA training: slow convergence and the degradation of discrete probability knowledge within pretrained VLMs. To mitigate these problems, we opt to first pretrain the VLM backbone using a cross-entropy loss before diffusion training, which is consistent with its original training objective. This helps us effectively preserve the

model's valuable pretrained knowledge, which additionally benefits from our add-ons of vision-language data during training. As illustrated in Fig. 6, our experiments confirm that this discretized pretraining phase significantly accelerates the convergence of the VLA model compared to training directly with the diffusion loss from the outset. In the following, we elaborate on the training details.

**RVQ Tokenizer.** To facilitate the cross-entropy training, we employ the RVQ (Van Den Oord et al., 2017; Esser et al., 2021; Lee et al., 2022) for discretization due to its high compression efficiency. Specifically, we first encode the continuous action chunk $\mathbf{A}_t \in \mathbb{R}^{T_a \times d}$ with a 1D temproal convolutional nerual networks (CNNs) $\phi_{\text{enc}}$ into $n$ latents of $C$ dimensions, denoted by $\{\mathbf{z}_i \in \mathbb{R}^C\}_{i=1}^n = \phi_{\text{enc}}(\mathbf{A}_t)$. For each $\mathbf{z}_i, 1 \leq i \leq n$, starting with $\mathbf{r}_0^i = \mathbf{z}_i$, we quantize it via an iterative process of depth $m$:

$$
\begin{aligned}
k_j^i &= \arg\min_{1 \leq k \leq K} \|\mathbf{r}_{j-1}^i - \mathbf{e}_j(k)\|_2^2, \\
\mathbf{r}_j^i &= \mathbf{r}_{j-1}^i - \mathbf{e}_j(k_j^i),
\end{aligned}
\tag{1}
$$

for $j = 1, \ldots, m$, where $\mathbf{e}_j \in \mathbb{R}^{K \times C}$ is the learnable codebook of size $K$ at depth $j$. As a result, $\{k_1^i, \ldots, k_m^i\}_{i=1}^n$ will be the token index for $\mathbf{A}_t$ and $\hat{\mathbf{A}}_t := \phi_{\text{dec}}(\{\hat{\mathbf{z}}_i\}_{i=1}^n) = \phi_{\text{dec}}(\{\sum_{j=1}^m \mathbf{e}_j(k_j^i)\}_{i=1}^n)$ will be the quantization result, where $\phi_{\text{dec}}(\cdot)$ is the reverse 1D CNN decoder. We minimize the following loss to train the tokenizer:

$$
\begin{aligned}
\mathcal{L}_{\text{vq}} := \mathbb{E}_{\{\cdot,\cdot,\mathbf{A}_t\} \sim \mathcal{D}, 1 \leq i \leq n} \Big[ &\|\mathbf{A}_t - \hat{\mathbf{A}}_t\|_2^2 \\
&+ \|\operatorname{sg}(\mathbf{z}_i) - \hat{\mathbf{z}}_i\|_2^2 + \beta\|\mathbf{z}_i - \operatorname{sg}(\hat{\mathbf{z}}_i)\|_2^2 \Big].
\end{aligned}
\tag{2}
$$

To mitigate the notorious codebook collapse, we have taken several measures during RVQ training, including lower codebook dimension (Yu et al., 2021), replacing the Euclidean distance with cosine similarity in Eq. (1) (Yu et al., 2021), smoothing codebook updates via exponential moving average (EMA) (Razavi et al., 2019), and restarting inactive codebook entries every fixed period (Zeghidour et al., 2021). As shown in Fig. 8, at the same level of quantization errors, our RVQ can compress action chunks into fewer tokens, which could greatly accelerate the large VLA's convergence.

**Model Details.** We selected the 7B Qwen2.5-VL (Bai et al., 2025) as our VLA backbone, leveraging its extensive

*Figure 2.* A three-stage pipeline for training RDT2. In Stage 1, we pre-train a 7B VLM backbone with discretized action data for vision-language reasoning capabilities. Then, in Stage 2, we train a small diffusion action expert to generate continuous actions efficiently. For highly dynamic tasks, we introduce a third stage that distills the diffusion policy into a one-step generator, thereby enabling extremely rapid inference speed.

pre-training on large-scale vision-language corpora. We project various modalities to a unified latent space for learning: vision and language by Qwen encoder, actions by our RVQ model. We reserved the $1024$ least frequent entries in the vocabulary to represent these action tokens. The VLA model was trained for $128$K iterations on a composite dataset of our UMI dataset and a small subset of vision-language data, using a next-token prediction objective.

## 5.2. Stage 2

To enhance inference efficiency beyond that of autoregressive models, we introduce a second stage of training. In this stage, we freeze the pretrained VLA backbone from Stage 1 and train a dedicated action expert. This expert, a 400M parameter variant of RDT-1B (Liu et al., 2024), is optimized for speed by substituting Multi-Head Attention (MHA) (Vaswani et al., 2017) with Grouped Query Attention (GQA) (Ainslie et al., 2023). It generates continuous actions through a diffusion process, which is conditioned on natural language and image representations encoded by the frozen VLA backbone. Specifically, the action expert leverages cross-attention to incorporate the latent features from each layer of the VLA backbone.

**Flow-Matching Training.** The action expert is supervised by a flow-matching loss (Lipman et al., 2022):

$$
\begin{aligned}
\mathcal{L}_{\text{expert}}(\theta) := \mathbb{E}_{\{\ell, \mathbf{o}_t, \mathbf{A}_t\} \sim \mathcal{D}, \tau \sim \mathcal{U}(0,1)} \Big[ \| \\
\mathbf{v}_\theta(\tau, \mathbf{A}_t^\tau, \text{VLA}(\ell, \mathbf{o}_t)) - \mathbf{u}(\mathbf{A}_t^\tau \mid \mathbf{A}_t) \|_2^2 \Big],
\end{aligned} \quad (3)
$$

where $\tau$ is the flow-matching time step, $\mathbf{v}_\theta(\cdot)$ is the denoising network with trainable parameters $\theta$, and $\text{VLA}(\cdot)$ is the frozen VLA backbone. Here, we denote the noisy action chunk by $\mathbf{A}_t^\tau := (1 - \tau)\epsilon + \tau \mathbf{A}_t$, where $\epsilon \sim \mathcal{N}(\mathbf{0}, \mathbf{I})$ is a random Gaussian noise. The ground-truth velocity is given by: $\mathbf{u}(\mathbf{A}_t^\tau \mid \mathbf{A}_t) := \mathbf{A}_t - \epsilon$. During inference, we first sample a Gaussian noise vector: $\mathbf{A}_t^0 \sim \mathcal{N}(\mathbf{0}, \mathbf{I})$ and then denoise it to a clean action chunk:

$$
\mathbf{A}_t^{\tau + \delta\tau} = \mathbf{A}_t^\tau + \delta\tau \cdot \mathbf{v}_\theta(\tau, \mathbf{A}_t^\tau, \text{VLA}(\ell, \mathbf{o}_t)), \quad (4)
$$

from $\tau = 0$ to $\tau = 1$. In practice, we set the step size $\tau = 0.2$, corresponding to $5$ integration steps. Besides, we only calculate $\text{VLA}(\cdot)$ once since it stays invariant during integration. In our experiment, we randomly initialized the action expert and trained it for $66$K iterations on our UMI dataset, with the VLA backbone (trained in Stage 1) frozen.

## 5.3. Stage 3

The action generation process, as formulated in Eq. (4), necessitates five sequential forward passes through the denoising network for each action chunk, which imposes a considerable inference overhead. This latency presents a practical bottleneck for tasks with high dynamic requirements, such as playing table tennis. To overcome this limitation, we employ diffusion distillation (Salimans & Ho, 2022; Chen et al., 2023) to convert the expert policy trained in Stage 2 into a single-step generator. As illustrated in Fig. 7, this technique drastically reduces model latency, enabling our large-scale VLA to achieve a significantly faster inference speed than much smaller models.

**Diffusion Distillation.** For highly dynamic tasks, we distill the action expert into a single-step generator with parameters $\theta'$ using the following regression objective:

$$
\begin{aligned}
\mathcal{L}_{\text{distill}}(\theta') := \mathbb{E}_{\{\ell, \mathbf{o}_t, \cdot\} \sim \mathcal{D}, \mathbf{A}_t^0 \sim \mathcal{N}(\mathbf{0}, \mathbf{I})} \Big[ \| \\
\mathcal{F}(\mathbf{A}_t^0, \ell, \mathbf{o}_t; \theta) - G(\mathbf{A}_t^0, \ell, \mathbf{o}_t; \theta') \|_2^2 \Big],
\end{aligned} \quad (5)
$$

where $\mathcal{F}(\cdot)$ denotes the generation process in Eq. (4) and $G(\mathbf{A}_t^0, \ell, \mathbf{o}_t; \theta') := \mathbf{A}_t^0 + \mathbf{v}_{\theta'}(0, \mathbf{A}_t^0, \text{VLA}(\ell, \mathbf{o}_t))$ is the target single-step generator. It is noted that $\theta$ has been pretrained in Stage 2 and stays frozen in this stage, $\text{VLA}(\cdot)$ is also frozen, and $\theta'$ is trainable, initialized from $\theta$. Unlike previous distillation practices, $\mathcal{F}(\cdot)$ is computed on-the-fly during training, rather than pre-generated during data preparation. This approach offers a compelling advantage with acceptable computational overhead. Generating low-dimensional actions is remarkably efficient (with a few integration steps), in stark contrast to the image or video generation requiring up to hundreds of steps. At the same

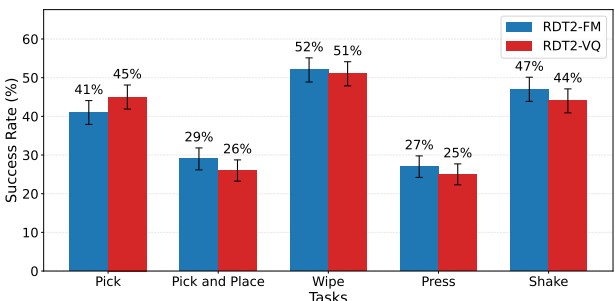

*Figure 3.* Results of zero-shot experiments of RDT2. The error bar represents the standard error.

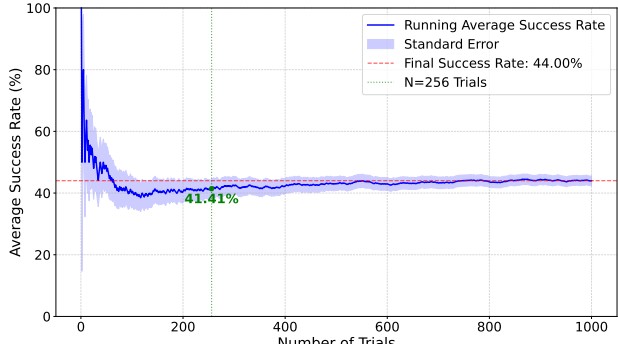

*Figure 4.* Convergence curve of statistical success rate in repeated trials (Pick Task, RDT2-FM).

time, it yields a significant benefit by substantially reducing the risk of the distilled policy overfitting to the pre-generated data, a common pitfall in regression-based distillation.

## 6. Experiments

This paper aims to achieve superior generalizability for robotic models by increasing the quantity and diversity of data, which will be rigorously verified in this section. While it is common practice to fine-tune VLAs before evaluation, we want to zero-shot test our RDT2 under "**4U**" setting — **U**nseen embodiment, **U**nseen scene, **U**nseen object, and **U**nseen instruction. For quantitative evaluation, we will conduct repeated experiments up to 1000 trials to ensure sufficiently low variance (see Fig. 4), which was previously lacking in many studies but is crucial for the reliability of results. To be specific, our experiments answer the following questions (see App. D for experimental details):

- $\mathcal{Q}1$: Can RDT2 effectively generalize to unseen embodiments, objects, scenes, and instructions, which is impractical for previous VLAs?
- $\mathcal{Q}2$: What is the scaling law of RDT2's generalizability with respect to training data and model size?
- $\mathcal{Q}3$: How does RDT2 compare to other VLAs, in terms of fine-tuning experiments on challenging dexterous, dynamic, or long-horizon tasks?
- $\mathcal{Q}4$: How does each component of our training strategy contribute to the performance of RDT2?

### 6.1. Zero-Shot Experiments

To answer $\mathcal{Q}1$, without any fine-tuning, we deployed RDT2 on unseen embodiments and evaluated it on tasks with unseen objects, scenes, and instructions. The model was pretrained solely on UMI human data and vision-language pairs, without any robotic data. We considered simple open-vocabulary tasks: pick up an object specified in free-form language, pick up a specified object and place it in a specified location, wipe a table with any cloth, press any button, and shake any object. The specific task settings are described in Fig. 15.

To ensure the rigor of the experiment, firstly, we selected three scenes that had never appeared in the training set for testing. These scenes are controllable and located in a laboratory with constant lighting, ensuring low variance and reproducibility of the results. Secondly, we purchased a new batch of objects for testing, ensuring these objects are unseen in the training set. Thirdly, to verify that the instructions are also unseen, we de-duplicated the test instructions according to the training set.

The results in Fig. 3 showed that both of our RDT2 variants could accomplish basic open-vocabulary tasks across combinations of unseen objects, scenes, instructions, and embodiments. Although the success rate is not high, the significance of this result is profound: large models trained solely on human data can achieve combinatorial generalization across multiple factors, including embodiment. Furthermore, we observed no significant difference between RDT2-VQ and RDT2-FM in terms of standard error. Combined with Fig. 7, this demonstrates that our Stage 2 training can improve the model's inference efficiency without any performance degradation.

To validate the reliability of our empirical success rate, we conducted $1,000$ trials on the Pick Task using the RDT2-FM model. As shown in Fig. 4, the success rate converged as the number of trials grew. And the region formed by the standard error always contained the final value (red dashed line). These proved the reliability of the experimental results. However, only with a sufficient number of trials could the standard error be reduced to an acceptable level. To balance between reliability and labor cost, we chose $n = 256$ trials for all subsequent experiments.

### 6.2. Scaling Laws of Data and Model Size

To answer $\mathcal{Q}2$ and precisely measure scaling behavior, we adopted the following experimental protocol: RDT2-VQ models of different sizes are each trained for one epoch on the full dataset, using uniform sampling. We evaluated the training loss at multiple intermediate checkpoints through-

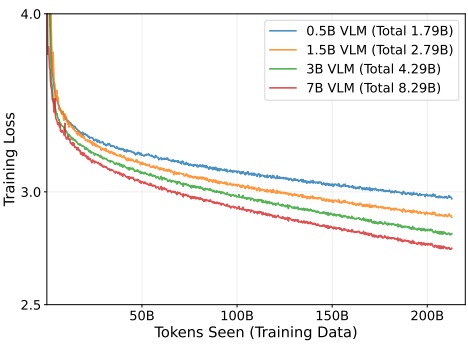 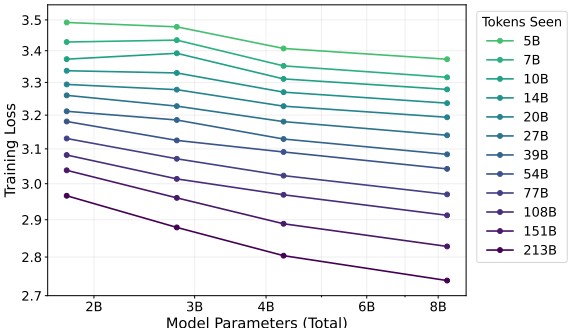

*Figure 5.* Scaling laws of RDT2. **Left**: Training loss as a function of consumed tokens (non-repeating) under various model parameter scales. "Total" parameters includes vision encoders. **Right**: Training loss as a function of total model parameters under different amounts of training data (measured by tokens).

out this single epoch. Since each data token was consumed only once, the training loss could indicate the model's generalizability on unseen samples. This design allows us to associate each checkpoint with an exact amount of effective compute ($C \propto N \times D$, where $N$ is the model size and $D$ is the number of data samples (i.e., tokens consumed) processed up to that point). Consequently, we could plot the training loss as a function of both model size ($N$) and tokens consumed ($D$), isolating their effects on performance.

Fig. 5 shows the scaling law curves of RDT2 which match the results from (Hoffmann et al., 2022; Kaplan et al., 2020):

$$\hat{L}(N, D) \triangleq E + \frac{A}{N^{\alpha}} + \frac{B}{D^{\beta}}, \qquad (6)$$

where the fitting results show $E \sim 2.1108, A \sim 4.3754 \times 10^3, \alpha \sim 0.4402, B \sim 1.7906 \times 10^2, \beta \sim 0.2251$.

This scaling law formula shows that increasing both model parameters and data scale leads to clear and consistent gains in model performance. It implies that identifying highly scalable data collection methods—such as data acquisition from wearable devices—and scaling up such data sources is

crucial for improving model intelligence.

### 6.3. Fine-Tuning Experiments

To answer $\mathcal{Q}$3, we compared RDT2 with the most advanced baselines: $\pi_0$-FAST (Pertsch et al., 2025) and $\pi_{0.5}$ (Intelligence et al., 2025). We finetuned each model for challenging real-world tasks, including long-horizon tasks (e.g., table bussing), deformable object manipulation (e.g., folding clothes, unzipping a zipper), and dynamic tasks (e.g., playing table tennis, rapid button pressing). We refer to Fig. 16 for task descriptions. We use the RDT2-UltraFast variant in this experiment.

As summarized in Tab. 2, RDT2 demonstrated superior performance across all task categories. Specifically, in deformable object manipulation, RDT2 achieved substantially higher success rates than baselines, particularly on the complex, multi-step cloth folding task. Notably, its performance on unseen objects was 4 times higher than the baseline, highlighting strong generalization. For the long-horizon table bussing task, RDT2 not only doubled the full-task success rate but also achieved a significantly higher average

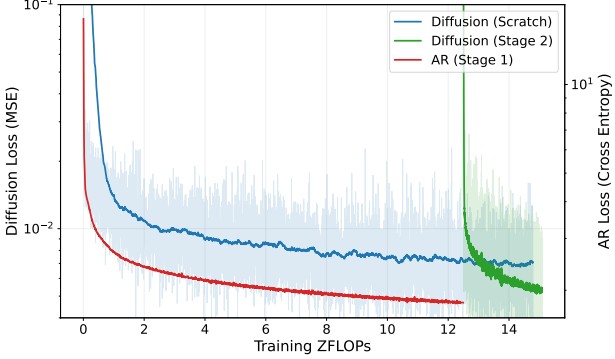 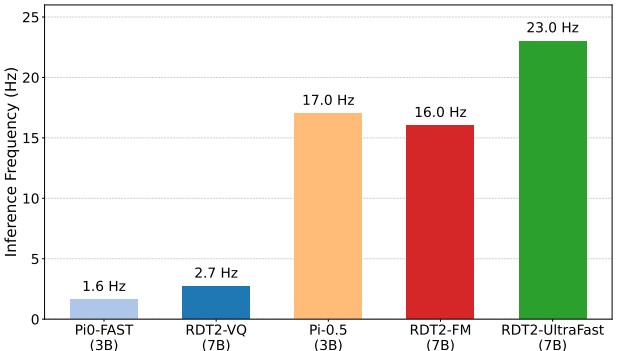

*Figure 6.* Loss curves of RDT2 (Diffusion vs. AR + Diffusion). AR + Diffusion achieves significantly faster convergence and lower loss. Diffusion loss is smoothed exponentially (99%). Shaded curves denote the raw data.

*Figure 7.* Comparison of inference frequency across various VLAs. Despite having a model size more than twice that of $\pi_{0.5}$, RDT2-UltraFast boasts the fastest inference speed.

*Table 2.* Fine-tuning performance of RDT2 and baseline models on challenging real-world tasks. The progress score is the average percentage of subtasks completed. For button pressing, we report the difference in reaction time between the policy and the human expert teleoperator (average 2661 ms). It is noted that the $\pi_0$-FAST model failed to produce a fast enough policy for playing table tennis.

| Task | Metric | RDT2 | $\pi_{0.5}$ | $\pi_0$-FAST |
|---|---|---|---|---|
| Cloth Folding | Success Rate(%) | **77** | 36 | 29 |
| | Subtask1: Left Sleeve(%) | **97** | 92 | 80 |
| | Subtask2: Right Sleeve(%) | **95** | 70 | 61 |
| | Subtask3: Final Fold(%) | **81** | 45 | 38 |
| | *Unseen* Object(%) | **51** | 15 | 10 |
| Table Bussing | Progress Score(max=1.0) | **0.58** | 0.39 | 0.30 |
| | *Unseen* Scene(max=1.0) | **0.33** | 0.17 | 0.11 |
| Unzipping | Success Rate(%) | **45** | 13 | 8 |
| Button Pressing | Reaction Time(ms) | **+97** | +323 | +981 |
| Table Tennis | Hit Rate(%) (1x/1.2x/1.5x/ 1.7x/2x speed) | **88/85/ 76/69/ 68** | 78/74/ 58/57/ 56 | *N/A* |

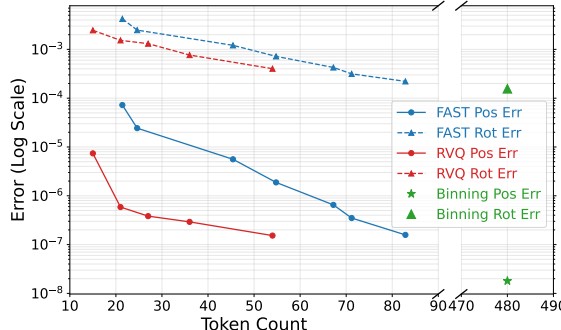

*Figure 8.* Discretization experiment: the position error (MSE) and rotation error (Radian) vs. the number of tokens in the discrete representation. At the same level of discretization error, our RVQ requires far fewer tokens.

due to fewer action tokens with the RVQ tokenizer. For diffusion, RDT2-UltraFast was the champion, thanks to the one-step diffusion distillation in Stage 3.

Fig. 8 evaluated the discretization error under different token budgets for representation. RVQ consistently achieved lower errors than the FAST tokenizer and saved up to about two-thirds of the tokens, because RVQ provided a more compact latent space for information compression. The uniform binning (Brohan et al., 2022; Zitkovich et al., 2023) achieved the lowest error but required far more tokens, rendering its in-efficiency.

## 7. Conclusion

In this work, we presented RDT2, a robotic foundation model designed to overcome the barriers of data scarcity, inference latency, and cross-embodiment generalization. By synergizing a massive, embodiment-agnostic dataset of over 10, 000 hours with a novel three-stage training strategy, we successfully bridged the gap between the discrete semantic reasoning of large VLMs and the continuous precision required for motor control. Our approach not only ensures real-time performance through effective distillation but also demonstrates unprecedented zero-shot transfer capabilities on novel objects, scenes, instructions, and even robotic platforms. Furthermore, in fine-tuning benchmarks, RDT2 also achieved state-of-the-art performance in dexterous, long-horizon, and dynamic tasks such as table tennis.

## Impact Statement

This work represents a significant step toward general-purpose embodied intelligence, potentially accelerating the deployment of robotic assistants in domestic and industrial settings, which could yield substantial benefits for elderly care and labor efficiency. However, the development of Vision-Language-Action (VLA) models trained on large-scale, real-world data introduces specific ethical consider-

progress score, indicating better long-horizon robustness. In dynamic tasks, thanks to distillation in Stage 2, RDT2 showed improved temporal responsiveness (faster button-press reaction time) and a higher ball-hitting rate in table tennis. In conclusion, the fine-tuning experiments confirmed that RDT2 effectively transfers its pre-trained knowledge to state-of-the-art performance in diverse challenging downstream applications.

### 6.4. Ablation Studies

To address $\mathcal{Q}4$, we conduct ablation studies on the key components of RDT2, including the hybrid training of auto-regression (AR) and diffusion (Stage 1 and 2), the RVQ for action discretization, and the distillation in Stage 3.

Fig. 6 compared diffusion-only training (we train both backbone and the action expert) with the proposed two-stage AR+Diffusion framework. AR pre-training avoided damaging discrete VLM knowledge and provided a good initialization, thus enabling faster convergence. Furthermore, we found the AR pre-training is also critical for achieving lower final loss.

Fig. 7 showed the inference speed of different baselines. For auto-regression, RDT2-VQ exhibited the highest frequency

ations. Primarily, the reliance on data collected from over 100 private households necessitates rigorous adherence to privacy standards and data anonymization to protect contributor identities. Furthermore, as RDT2 enables zero-shot deployment on novel robotic embodiments, it introduces physical safety risks associated with unpredictable behavior in unseen physical contexts; consequently, we emphasize that future deployment must be accompanied by robust safety guardrails and verification protocols to prevent harm in human-robot interaction scenarios.

## Acknowledgments

This work was supported by NSFC Projects (Nos.92570001, 62550004, 62522609, 92470118, 92370124, U25B6003, 62350080), the Beijing Natural Science Foundation (No. L247030); and the High Performance Computing Center, Tsinghua University.

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

# A. UMI Hardware Specifications

## A.1. Data Collection Hardware (Handheld UMI)

The handheld data collection device (Fig. 9a) integrates a computing unit, a high-frequency vision system, precise infrared tracking, and a custom gripper interface.

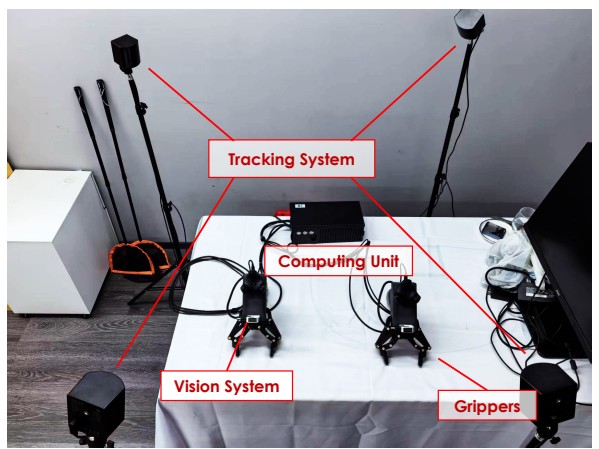

*(a)* Illustration of the Handheld Data Collection Device.

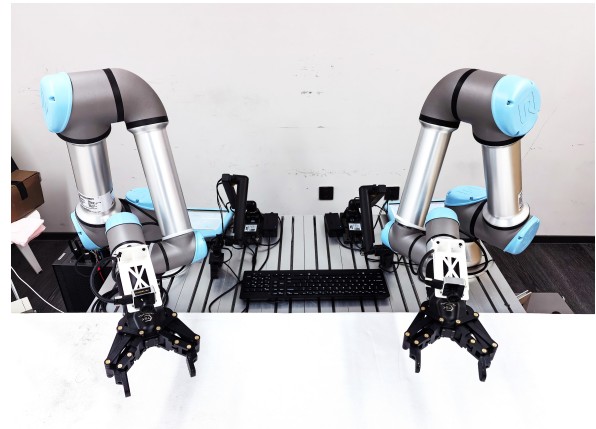

*(b)* Illustration of the Home Pose of the Arms.

*Figure 9.* Hardware Configuration for Data Collection and Deployment.

**Computing Unit.** Data logging is handled by an industrial control unit powered by an **Intel Core Ultra 7 155H** processor with 32GB RAM and 2TB SSD.

**Vision System.** We utilize the **Hikrobot MV-CS016-10UC** industrial camera (Hikrobot MV-CS016-10UC product page).

- Sensor: Sony IMX273 Global Shutter CMOS (1/2.9")

- Resolution: $1440 \times 1080$ (1.6 MP)

- Pixel Size: $3.45\mu m \times 3.45\mu m$

- Frame Rate: Up to 249 fps (configured to 30 Hz for collection)

- Interface: USB 3.0

**Tracking System.** We adopted an infrared light-based positioning system using 4 **HTC VIVE Tracker 3.0** (HTC VIVE Tracker 3.0 product page) to track the 6-DoF pose of the end-effector.

**Grippers.** To ensure consistent contact dynamics, as we employ the ZhiXing CTAG2F120 gripper (ZhiXing gripper product page) for robotic execution in the actual embodiment. For the handheld data collection device, we developed a custom non-actuated replica that preserves the exact configuration and geometry of the original ZhiXing gripper. The replica chassis is fabricated via CNC-machining from Nylon 66 reinforced with Glass Fiber (PA66+GF), utilizing stainless steel and copper for auxiliary components. This unit features a sliding rail mechanism equipped with mechanical limit switches, allowing the operator to manually control the gripper's opening width to strictly match the kinematic range of the robotic counterpart.

## A.2. Robotic Deployment Setup

We deploy our policy on two distinct robotic arms to evaluate cross-embodiment transfer: the **Franka Research 3 (FR3)** (FR3 product page) and the **Universal Robots UR5e** (UR5e product page).

Both robotic arms are equipped with the same ZhiXing parallel jaw gripper used in data collection. Visual feedback is provided by the Hikrobot MV-CS016-10UC camera mounted in an eye-in-hand configuration, identical to the handheld setup.

To align the inference and training state distributions, the robot is initialized to a home pose (Fig. 9b) that visually replicates the average starting perspective of the handheld data collection device.

## B. UMI Dataset

### B.1. Dataset Overview

The UMI dataset contains approximately 10,000 hours of interaction data collected in over 100 unique home environments. While the dataset includes data from structured settings such as showrooms, mock-up apartments, restrooms, and nursing homes, a significant component consists of data collected in private homes via paid crowdsourcing.

Data collection in residential environments captures long-tail object categories, diverse materials, and spatial arrangements that are difficult to replicate in controlled laboratories. These features improve the dataset's ecological validity and support the learning of representations that generalize across environments and deployment contexts.

Recent large manipulation datasets have similarly focused on scale and diversity. The DROID dataset(Khazatsky et al., 2024) includes 76,000 teleoperated trajectories from hundreds of indoor scenes, indicating that diverse real-world data improves generalization. FastUMI-100K(Zhaxizhuoma et al., 2025; Liu et al., 2025) contains over 100,000 trajectories from household environments with 50 tasks and hundreds of objects, showing that large-scale multimodal data enhances policy performance. Consistent with these findings, the UMI dataset highlights scale, scene variety, and task coverage as essential for representation learning and generalization in robot manipulation.

### B.2. Data Collection in In-Home Environments

In home environments, we defined over 50 tasks that cover various daily manipulation activities (Fig. 10), such as picking and placing, pouring, wiping, shaking, stirring, and organizing. In each recording session, data collectors were given a high-level instruction (see Table 3) and performed the task using objects found in their homes.

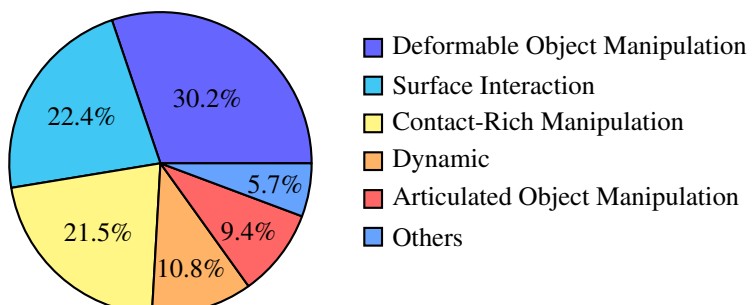

*Figure 10.* Distribution of In-Home Manipulation Tasks.

The data collection protocol is explicitly designed to promote diversity. We encouraged collectors to vary their choice of objects, containers, and strategies. For example, in packing tasks, collectors may place items into backpacks, plastic bags, or baskets depending on availability. These in-home recordings include interactions with over 1,000 unique objects.

The dataset also includes contact-rich interactions such as pressing, plugging, and operating doors, as well as deformable object manipulation like folding clothes and cleaning. We also included long-horizon tasks, such as organizing cluttered surfaces, which require multi-step planning and continued execution.

### B.3. Facility-Based Collection of Core Manipulation Primitives

To supplement the in-home data, we collected manipulation data in a dedicated facility designed for parallel data collection. The facility has 50 workstations, so multiple collectors can record simultaneously under controlled layout and sensing conditions. This setup ensures consistent coverage of core manipulation skills.

| ID | Task Type | Instruction | Requirements |
|---|---|---|---|
| A038 | Wiping | Clean diverse household surfaces using a slightly damp cloth for a fixed duration (20 min). Vary materials, object categories, motion angles, speeds, and cloth types. Prepare all objects beforehand and retry failures. | Sustained contact control, motion diversity, force modulation, temporal consistency |
| A039 | Picking | Collect varied kitchen waste items and place them into a garbage bag for 20 minutes. Maximize variation in object types, sizes, weights, and locations. Use diverse grasping strategies. Avoid damaging materials. | Grasp diversity, spatial search, clutter handling, object transfer robustness |
| A040 | Organizing | Organize utensils, cookware, condiments, and tools into proper locations for 30 minutes. Includes opening and closing drawers or cabinets. Use many object categories and storage positions. Prepare items in advance and retry failures. | Multi-step planning, object categorization, articulated object interaction, sequential manipulation |

*Table 3.* Representative Instruction Styles for In-Home Tasks

We used a pool of over 3,000 objects with varying shapes, sizes, weights, materials, and everyday categories (e.g., containers, tools, packages, and deformable items). By sampling object combinations across stations, we ensured diversity in grasping and contact interactions while maintaining a consistent task structure.

Instructions in this setting focused on manipulation primitives like grasping, lifting, moving to a target region. This controlled data complements the in-home dataset by reinforcing low-level manipulation representations in a reproducible setup.

### B.4. Two-Stage Annotation Pipeline

We annotated the UMI dataset using a two-stage pipeline for both human and machine annotation (Fig. 11).

First, recordings are segmented based on high-level task instructions to create task-level clips. Second, these clips are decomposed into fine-grained action segments. Annotations are stored in natural language (e.g., "Grasp the red cup using the right hand") but follow a structured schema specifying the hand, object, and action primitive. This ensures grounding between perception, language, and motor behavior.

### B.5. Language Augmentation

To improve linguistic coverage, we applied systematic language augmentation to the fine-grained annotations. For each instruction, we generated semantically equivalent paraphrases and simplified variants that omit specific hand or object details. For example, the instruction *"Put the black-handled rolling knife on the near left side of the table using the left hand"* was rewritten as *"Using your left hand, place the rolling knife with the black handle onto the near left side of the table"* or simplified to *"Place the knife on the left."* This process enhances robustness to linguistic variation and supports generalization across instruction styles. Machine annotations were generated using Google Gemini 2.5 Pro(Google Gemini 2.5 Pro Documents). We show the examples of our augmented language in Fig. 12.

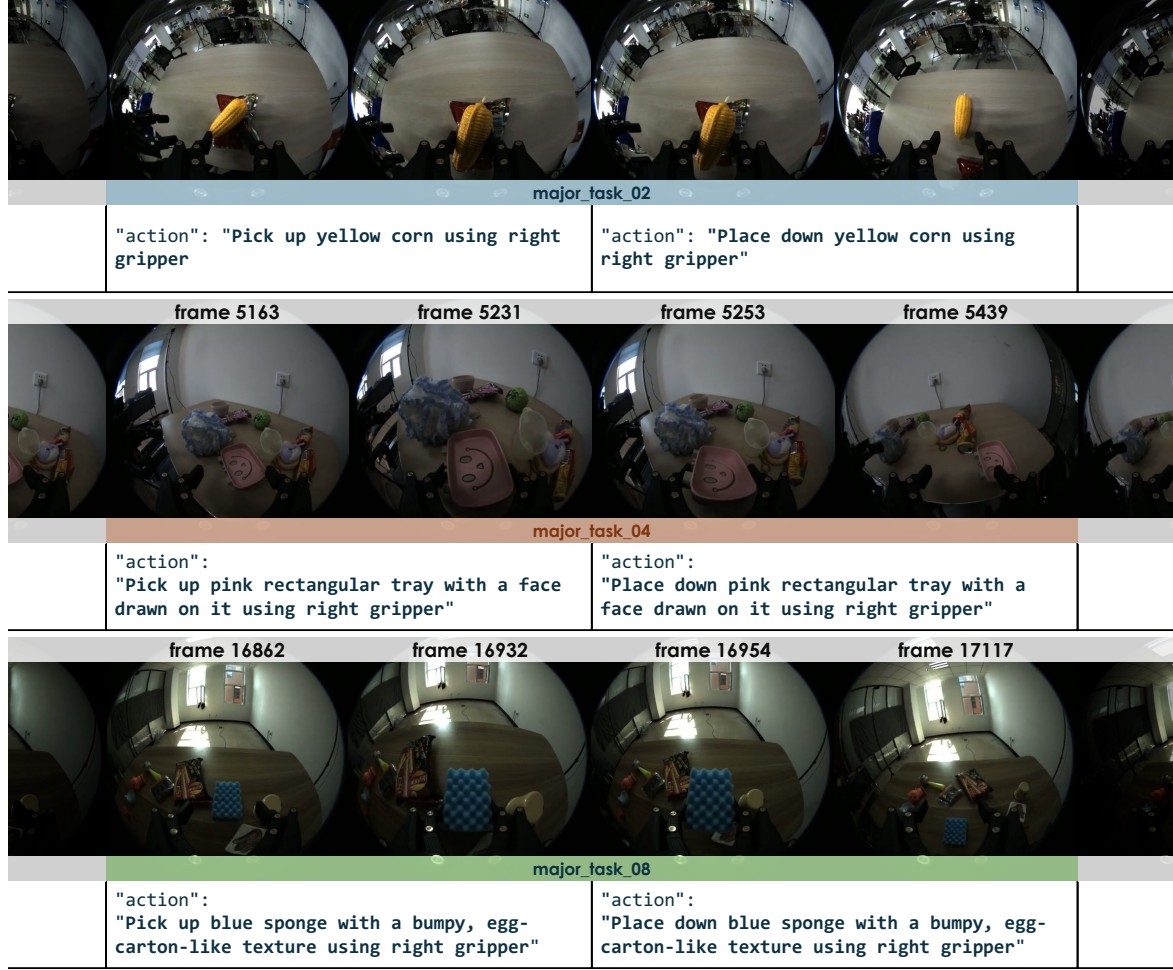

*Figure 11.* Annotation examples.

### B.6. Vision–Language Question Answering Pre-training Datasets

Our VLA model was pretrained on a collection of egocentric and robotics-relevant visual question answering (VQA) datasets, containing over 12 million question–answer pairs. This corpus combines Internet-scale vision–language data with embodied QA datasets, covering static images, egocentric videos, and robot manipulation scenarios. This provides supervision for semantic grounding, temporal reasoning, spatial understanding, and language–action alignment.

- **Ego4D + QaEgo4D.** Ego4D is a massive egocentric video dataset and benchmark suite collected across thousands of hours of daily-life first-person video, including natural language query tasks designed to probe episodic memory, temporal localization, and semantic understanding in long video sequences (Grauman et al., 2021). Extensions such as QaEgo4D build on these annotations to provide explicit visual QA pairs from the egocentric video streams (Bärmann & Waibel, 2022).

- **HD-EPIC.** The HD-EPIC dataset extends egocentric video understanding to highly detailed kitchen environments with dense annotations, including multiple types of VQA questions that require fine-grained action recognition, object motion comprehension, and 3D spatial reasoning over long first-person video clips (Perrett et al., 2025).

- **RoboVQA.** RoboVQA is a large multimodal QA benchmark designed for robotic reasoning over long-horizon video data. It contains hundreds of thousands of question–answer pairs drawn from robot and tool embodiment scenarios and is suited for affordance reasoning and future prediction tasks (Sermanet et al., 2023).

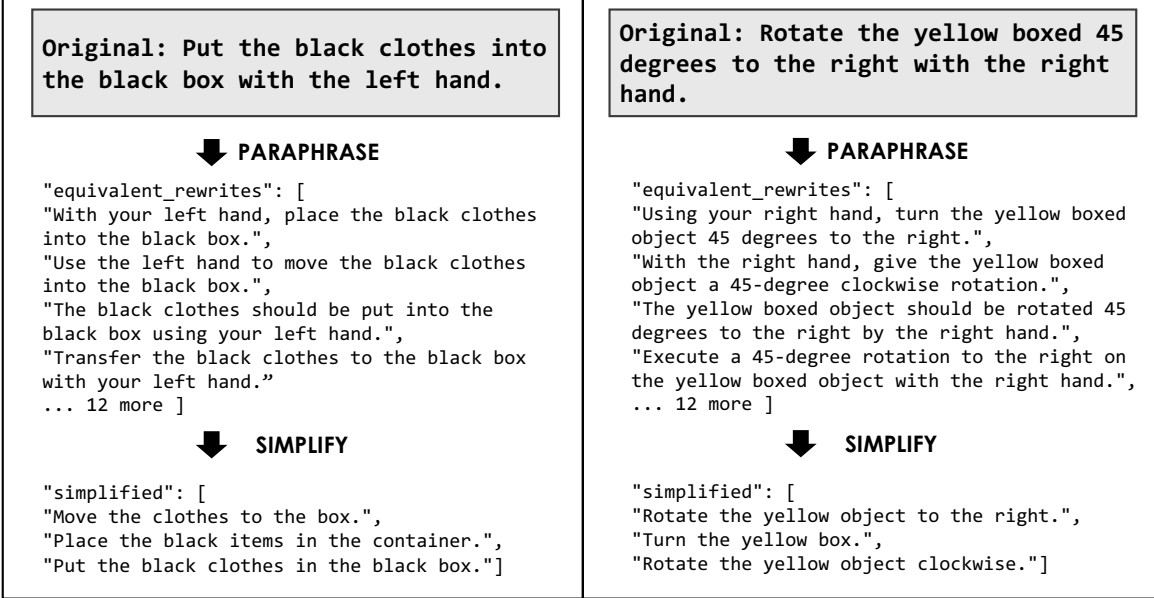

*Figure 12.* Illustration of Language Augmentation.

- **RoboBrain (ShareRobot).** The ShareRobot dataset, introduced as part of the RoboBrain framework, comprises over one million question–answer pairs annotated with task planning, affordance, and trajectory information across diverse robotic manipulation episodes, enabling models to learn structured planning and action reasoning (Ji et al., 2025).

- **Other Datasets.** The remaining data sources include large Internet-scale vision–language collections such as PixMo-Cap-QA and Cambrian-10M, which provide broad visual and linguistic coverage to strengthen general semantic alignment and instruction understanding in multimodal pretraining (Deitke et al., 2024; Tong et al., 2024).

## C. Training Details

### C.1. Platform and Data Pipeline

We implement our training framework using PyTorch (link to PyTorch codebase) and DeepSpeed (link to DeepSpeed codebase) to facilitate efficient distributed training. A core component of our infrastructure is the use of high-throughput WebDataset (link to WebDataset codebase) streaming. We convert all datasets into POSIX `tar` shards and utilize the `Resample` mode, enabling infinite data streaming without epoch boundaries. Data from heterogeneous sources is dynamically blended during training using `wds.RandomMix`, allowing us to adjust the sampling weights of different datasets on the fly.

### C.2. Stage 1: VQ Pretraining

In the first stage, we align the Qwen2.5-VL backbone with the robotic domain. The model is trained to predict discretized action tokens using a standard cross-entropy objective.

To ensure robust visual representations, we apply a comprehensive suite of image augmentations. We utilize standard color jittering (brightness, contrast, saturation, hue) alongside a randomized chain of image corruptions, including Gaussian/Laplace noise injection, motion blur, and JPEG compression artifacts.

We employed a cosine learning rate scheduler during training, which was additionally annealed with exponential decay over the final 8K iterations.

### C.3. Stage 2: Continuous Action Expert

In the second stage, we freeze the VQA backbone and optimize the RDT action expert using Conditional Flow Matching (CFM).

| Model Component | Layers | Hidden Size | Heads | KV Heads | Parameters |
|---|---|---|---|---|---|
| Qwen2.5-VL Backbone | 28 | 3584 | 28 | 4 | ∼7B |
| RDT2 Action Expert | 14 | 1024 | 8 | 4 | ∼400M |

*Table 9.* RDT2 model configuration.

| Hyperparameter | Stage 1 (VQ Pretraining) | Stage 2 (Flow Matching) |
|---|---|---|
| Batch Size (Per-GPU) | 96 | 96 |
| Learning Rate | $1 \times 10^{-4}$ | $1 \times 10^{-4}$ |
| LR Schedule | Cosine Decay | Constant |
| Warm-Up Steps | 1000 | 500 |
| Optimizer | AdamW | AdamW |
| $\beta_1, \beta_2$ | 0.9, 0.999 | 0.9, 0.999 |
| Weight Decay | $1 \times 10^{-2}$ | $1 \times 10^{-2}$ |
| $\epsilon$ | $1 \times 10^{-8}$ | $1 \times 10^{-8}$ |
| Mixed Precision | BFloat16 | BFloat16 |
| Gradient Clipping | 1 | 1 |
| Timestep Sampling | – | Logistic Normal ($\mu = 0, \sigma = 1$) |

*Table 10.* RDT2 training hyperparameters.

**Training Configuration.** We use a Logistic Normal distribution ($\mu = 0, \sigma = 1$) to sample timesteps $t$ during training, rather than a uniform distribution. This empirically concentrates the training budget on the most complex regions of the flow trajectory ($t \approx 0.5$). To monitor convergence, we evaluate the model every 2,500 steps using full multi-step integration on a held-out validation set. We report Action MSE, end-effector Position MSE, Rotation Geodesic Error, and Gripper Width MSE, and select checkpoints based on the aggregated validation error.

### C.4. Stage 3: One-Step Distillation

To enable high-frequency inference, we distill the Stage 2 policy into a single-step generator. We freeze the Stage 2 model as a *teacher* and train a *student* copy to regress the teacher's multi-step output in a single forward pass. The student is conditioned on $t = 0$ and minimizes the mean squared error (MSE) between its predicted velocity and the teacher's effective trajectory.

### C.5. Model and Training Configuration

We summarize the complete model architecture and training hyperparameters in Table 9 and Table 10, respectively.

## D. Experiments

### D.1. Baseline Implementations

We compare RDT2 against two baseline models: $\pi_{0.5}$ and $\pi_0$-FAST(Black et al., 2024; Pertsch et al., 2025; Intelligence et al., 2025). We implemented both baselines using the official OpenPI codebase (link to OpenPI codebase). We did not modify the architecture, only the configuration files and checkpoint paths to ensure fair comparison.

For $\pi_{0.5}$, we used the standard flow-based formulation. We trained the model for 20,000 steps on one node with 8 GPUs, with a batch size per GPU of 32. The configuration followed the official $\pi_{0.5}$ setup, with a discrete state input, an action horizon of 24, and a 32-dimensional action space. We initialized the action expert from Gemma-300M weights and the language backbone from Gemma-2B. Training used bfloat16 precision, AdamW optimization, and gradient clipping of 1.0.

For $\pi_0$-FAST, we trained the FAST tokenizer on the same RVQ action data used by the policy, following the standard FAST procedure. After training, we fixed the tokenizer and used it for policy training. We trained the policy for 30,000 steps on one node with 8 GPUs, with a batch size per GPU of 32. Other optimizer and scheduler settings matched those for $\pi_{0.5}$.

We trained each baseline model until stable convergence for each task (see Fig. 13 and Fig. 14). Table 11 details the training resources, and Table 12 lists the hyperparameters.

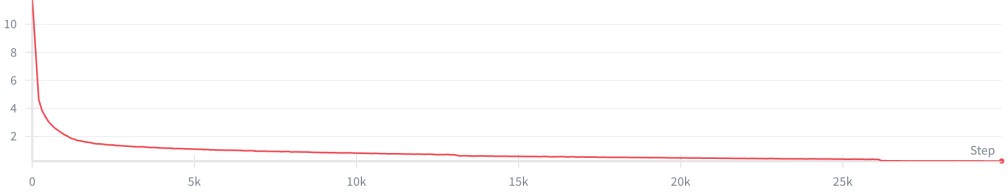

*Figure 13.* Loss of $\pi_0$-FAST on Table Bussing Task

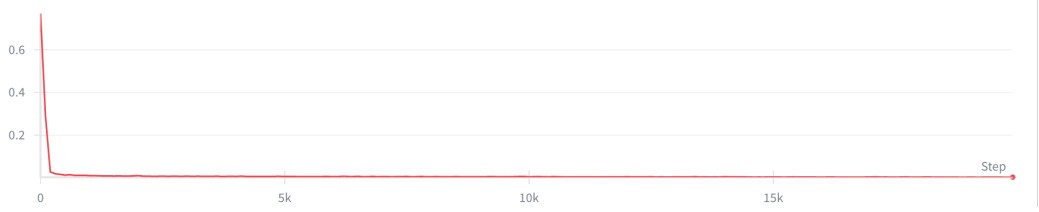

*Figure 14.* Loss of $\pi_0$.5 on Table Bussing Task

### D.2. Implementation and Model Configuration of RDT2

**Zero-Shot Configuration.** In the zero-shot experiments, we evaluate the generalizability of the pre-trained model directly on unseen tasks without any further updates. We utilize the model weights obtained from Stage 1 (for the RDT2-VQ variant, trained on the UMI dataset for 128k steps) or Stage 2 (for the RDT2-FM variant, action expert trained on the UMI dataset for 66k steps), which were trained solely on the large-scale UMI dataset. No task-specific data is involved in this setting, allowing us to assess the model's capability to handle novel instructions, objects, and scenes solely based on its pre-training knowledge.

**Fine-Tuning Configuration.** We initialized all fine-tuning experiments from a Stage 1 checkpoint pretrained on the UMI dataset for 128K steps. The vision-language backbone remained frozen. We considered two variants: RDT2-FM and RDT2-UltraFast. We fine-tuned both variants independently for each downstream task.

- **RDT2-FM**: We fine-tuned this variant using the flow matching objective. We trained the model for 50K steps on one node with 8 GPUs, using a global batch size of 96. Hyperparameters were consistent with Stage 2 training (Table 13).

- **RDT2-UltraFast**: To obtain this variant, we started with the fine-tuned RDT2-FM model. We then performed consistency distillation to convert it into a one-step generator. Distillation lasted for 20K steps, using the same hardware (1 node, 8 GPUs) and batch size (96).

We report the results of RDT2-UltraFast in fine-tuning experiments.

### D.3. Inference Configuration

For deployment and real-robot evaluations, we adjust the action chunk size to $T_a = 32$. Visual input is provided via a stereo setup using two cameras. The state dimension is fixed at 14, which is mapped from the original proprioceptive representation, where necessary to maintain consistency across different robotic platforms. All experiments are conducted for 256 trials.

### D.4. Task Descriptions

We describe the tasks used in our Zero-Shot and Fine-Tuning experiments below. All tasks took place in real-world environments.

#### D.4.1. ZERO-SHOT TASKS

In the Zero-Shot setting (Fig. 15), we evaluated generalization to unseen conditions. We adopted a "**4U**" protocol: **U**nseen embodiment, **U**nseen scene, **U**nseen object, and **U**nseen instruction. We conducted experiments across **3 environments** and

| Method | Steps | GPUs | Batch Size | Precision |
|---|---|---|---|---|
| $\pi_{0.5}$ | 20K | $1 \times 8$ | $32 \times 8$ | bf16 |
| $\pi_0$-FAST | 30K | $1 \times 8$ | $32 \times 8$ | bf16 |

*Table 11.* Baseline training configurations.

| Hyper-Parameter | Value |
|---|---|
| Optimizer | AdamW |
| Learning Rate | $2.5 \times 10^{-5}$ |
| Warm-Up Steps | 1,000 |
| $\beta_1, \beta_2$ | 0.9, 0.95 |
| Weight Decay | $1 \times 10^{-10}$ |
| Gradient Clipping | 1 |
| Mixed Precision | bf16 |

*Table 12.* Optimization hyper-parameters for $\pi_0$ and $\pi_{0.5}$.

**2 embodiments** (Franka Research 3 and UR5e), using **over 100 unseen objects**. We applied language augmentation to the prompts to ensure the model encountered **unseen instructions**. Each task was evaluated over **256 trials**. We designed five primitives to test manipulation capability.

**1. Pick Task**   This task evaluates the ability to identify and grasp a target object specified by natural language.
   **Setup**: **6 random objects** are placed in a cluttered arrangement. Positions and orientations are randomized.
   **Objects**: We use varying household objects with diverge geometries and texturesfrom a pool of over 100 unseen items.
   **Instruction**: Natural language instructions such as "Pick up the red apple using the right hand".
   **Success Metric**: A trial is successful if the robot grasps the correct object and lifts it at least 10cm without dropping it for 3 seconds.

**2. Pick & Place Task**   This task requires transporting a grasped object to a location.
   **Setup**: A chaotic scene with **5 to 10 objects** and a target container. Arrangements are randomized.
   **Instruction**: Two-stage instructions specifying the object and destination, e.g., "Pick up the banana... Put the banana in the silver bowl".
   **Success Metric**: Success requires picking the correct object and placing it in the container.

**3. Wiping Task**   This task tests the robot's ability to manipulate a tool (a towel) to interact with a surface or object, requiring sustained contact and motion control.
   **Setup**: A **towel** is placed on the table. We utilize a collection of ~**15 different types of towels** with varying textures and sizes. The robot must grasp the towel and perform a wiping motion on either the table surface or **an object (e.g., a bowl)**, as specified.
   **Instruction**: Instructions involve two phases, e.g., "Pick up the towel. Wipe the table with the towel" or "Pick up the towel. Clean the bowl with the towel".
   **Success Metric**: The trial is considered successful if the robot picks up the towel and performs a clear wiping action on the target surface.

**4. Shaking Task**   This task assesses the understanding of dynamic actions and object properties.
   **Setup**: A bottle is placed on the table. We use ~**20 different types of bottles**.
   **Instruction**: Instructions involve two phases, e.g., "Pick up the bottle. Shake the bottle".
   **Success Metric**: Success is defined by the robot grasping the object and performing a clearly visible shaking motion.

**5. Button Pressing**   This task evaluates precise positioning and force application on small targets.
   **Setup**: A keyboard is placed on the table. We use ~**10 different types of keyboards**.
   **Instruction**: "Press any key" or "Hit the keyboard".
   **Success Metric**: The trial is successful if the robot's end-effector presses any key on the keyboard.

| Hyper-Parameter | Value |
|---|---|
| Batch Size | 96 |
| Learning Rate | $1 \times 10^{-4}$ |
| Optimizer | AdamW |
| $\beta_1, \beta_2$ | 0.9, 0.999 |
| Weight Decay | $1 \times 10^{-2}$ |
| $\epsilon$ | $1 \times 10^{-8}$ |
| Gradient Clipping | 1 |
| Mixed Precision | bf16 |

*Table 13.* RDT2 finetuning hyper-parameters.

### D.4.2. FINE-TUNING TASKS

For fine-tuning experiments (Fig. 16), we selected tasks requiring dexterity, long-horizon planning, or dynamic control. We fine-tuned the model on 200 demonstrations for each task.

**1. Cloth Folding**  This is a highly challenging deformable object manipulation task involving a sequence of precise actions.
   **Task Goal**: Fold a long-sleeved shirt placed flat on the table into a compact square.
   **Procedure**: The task implies three distinct subtasks: 1) **Left Sleeve**: Fold the left sleeve inwards; 2) **Right Sleeve**: Fold the right sleeve inwards; 3) **Final Fold**: Fold the bottom of the shirt upwards to the neck.
   **Evaluation**: The total success rate is calculated based on the successful completion of all three subtasks.
   **Unseen Object Setting**: We evaluate on **3 different types of unseen shirts** featuring varying colors, textures, and sizes compared to the fine-tuning data to test generalization.

**2. Table Bussing (Long-Horizon)**  A task requiring sequential manipulation of multiple objects to prepare a dining setup.
   **Task Goal**: return all items (e.g., a plate, a cup, and cutlery) from random initial positions to the correct locations to form a dining setup.
   **Complexity**: This is a long-horizon task that requires the robot to plan and execute multiple pick-and-place actions in sequence. The order of operations may vary, and the robot must handle clutter.
   **Metric - Progress Score**: We use a Progress Score to evaluate performance. Each item picked up and successfully placed into the designated place contributes **0.2 points** to the score. Points are only awarded if the item is correctly placed in the designated location.
   **Unseen Scene**: We modify the table background and lighting conditions, testing on **2 distinct unseen scenes** to evaluate robustness to visual domain shifts.

**3. Unzipping**  Another deformable object task requiring fine motor skills.
   **Task Goal**: Unzip a zipper on a bag.
   **Setup**: The task involves bimanual manipulation where **both the left and right hands participate**. The robot must grasp the small fabric doll attached to the zipper tab and pull it along a specific trajectory.
   **Success** : A trial is considered successful only if the zipper is successfully unzipped.

**4. Button Pressing (Dynamic)**  Unlike the static zero-shot version, this task focuses on reaction speed.
   **Setup**: A keyboard is placed on the table, and a screen is positioned in front of the robot. The screen turns green at a random time.
   **Task Goal**: The robot must press any key on the keyboard as quickly as possible after the screen turns green.
   **Metric - Reaction Time**: We measure the time interval between the screen turning green and the key pressing. If the robot presses a key before the screen turns green, the trial is not counted as a success.

**5. Table Tennis**  A highly dynamic task requiring rapid visual processing and motion generation.
   **Setup**: A ball launcher shoots a ping-pong ball towards the robot. The robot holds a paddle.
   **Task Goal**: The robot must intercept and hit the moving ball.
   **Metric - Hit Rate**: The percentage of balls successfully hit by the paddle. This tasks effectively benchmarks the inference latency and control frequency of the policy, as slow models will consistently miss the ball.

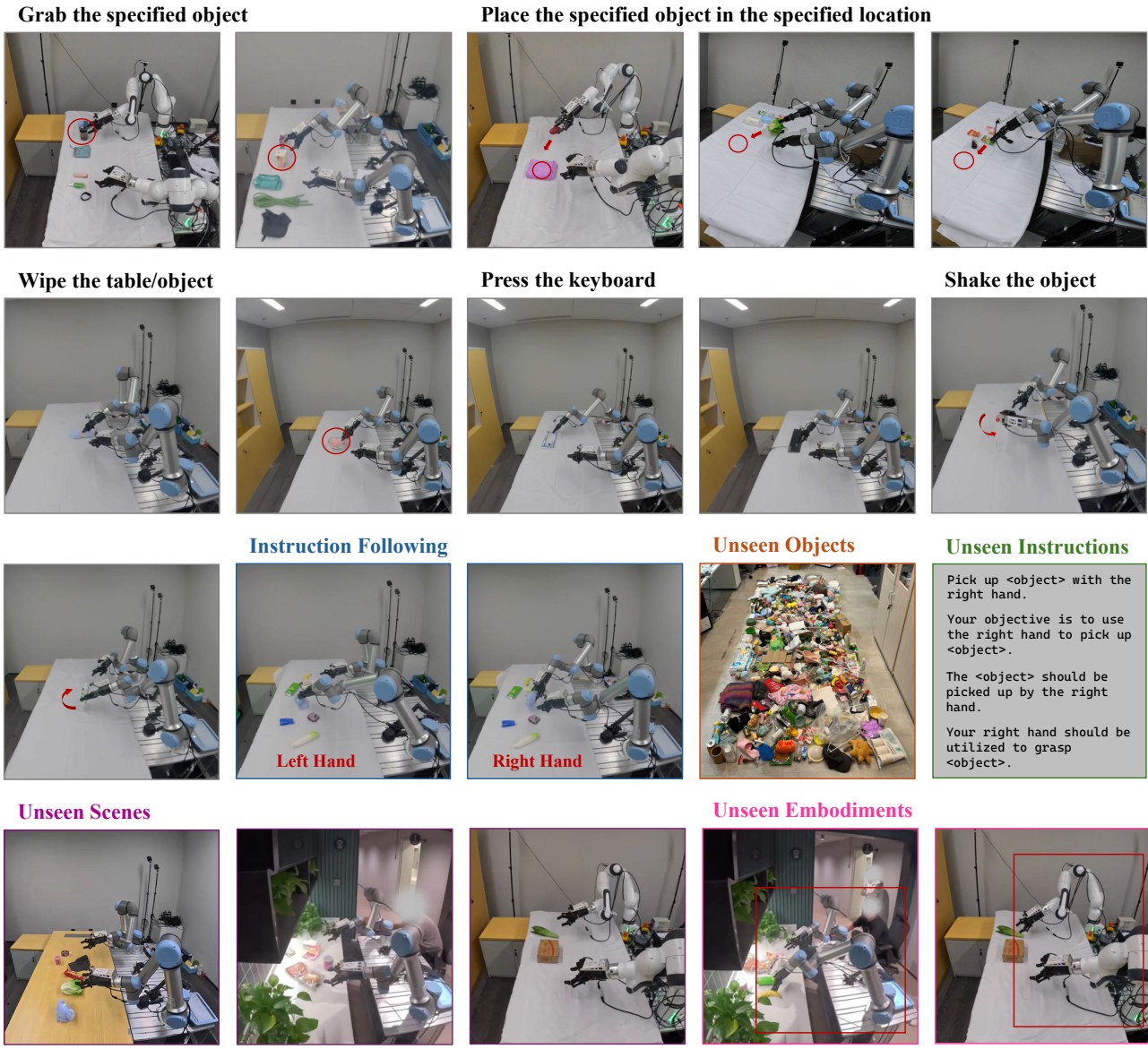

*Figure 15.* Demonstrations of zero-shot experiments of RDT2.

## D.5. Implementation Details of Ablation Studies

To test our hybrid training strategy (Stage 1 + Stage 2), we compared it against training the Action Expert from scratch with only Flow Matching.

**Hybrid Training (Stage 1: AR Pre-training).** We used the standard Cross-Entropy objective to pre-train the model, minimizing the negative log-likelihood of discretized action tokens:

$$\mathcal{L}_{\text{AR}} = -\sum_{t=1}^{T} \log P(a_t | a_{<t}, V, L), \tag{7}$$

where $a_t$ is the action token at step $t$, and $V, L$ are visual and language inputs. Configuration followed Table 10. We trained for $128$K steps on 7 nodes (8 GPUs each). We used a batch size per GPU of 96 with gradient accumulation, for a global batch size of $5,376$.

**Task 1: Cloth Folding**

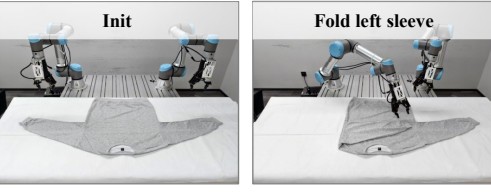 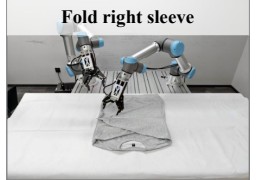 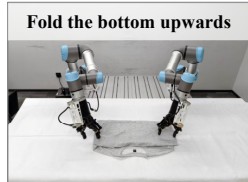 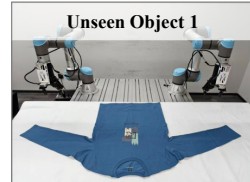

**Task 2: Table Bussing**

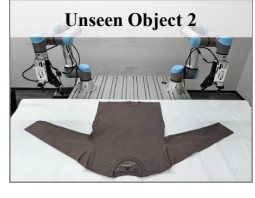 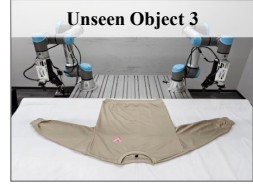 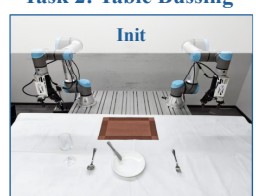 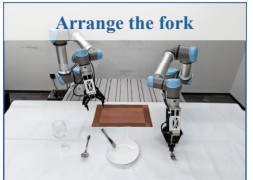 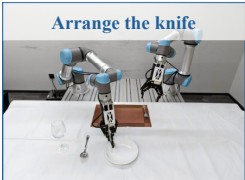

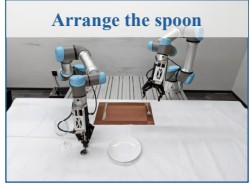 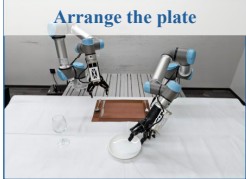 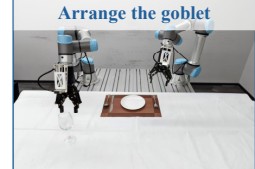 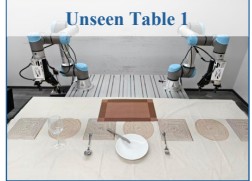 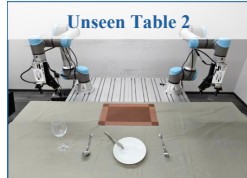

**Task 3: Unzipping**    **Task 4: Button Pressing**

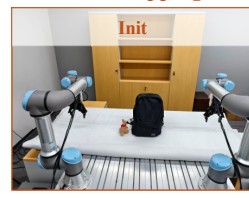 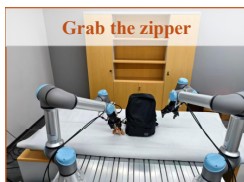 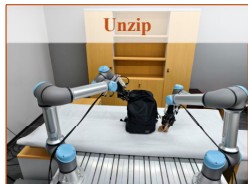 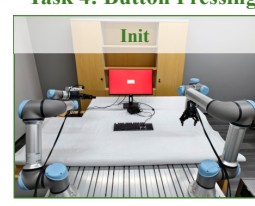 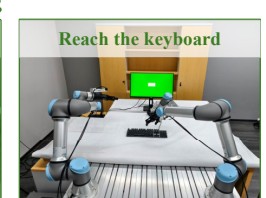

**Task 5: Table Tennis**

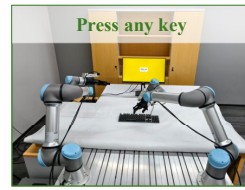 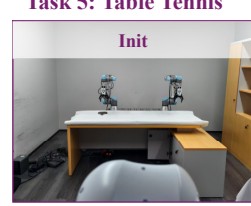 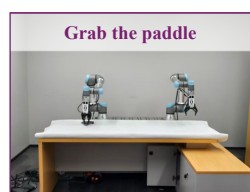 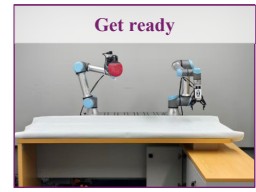 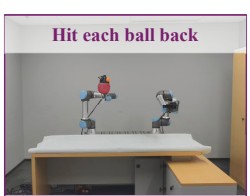

*Figure 16.* Demonstrations of fine-tuning experiments of RDT2.

**Hybrid Training (Stage 2: Diffusion Fine-tuning).** We froze the vision-language backbone and fine-tuned the Action Expert with Conditional Flow Matching (CFM). We minimized the flow matching loss:

$$\mathcal{L}_{\text{CFM}} = \mathbb{E}_{t,x_0,x_1} \|v_t(x_t) - (x_1 - x_0)\|^2. \tag{8}$$

Parameters are in Tab. 10. We trained for 66K steps on 7 nodes (8 GPUs each) with a batch size per GPU of 96, for a global batch size of 2,304.

**Diffusion from Scratch.** We trained the entire model (including the VLM backbone) from scratch using the same CFM loss. To match the scale of the hybrid training, we trained for 187K steps on 7 nodes (8 GPUs each). We used a batch size per GPU of 64 (via gradient accumulation), for a global batch size of 3,584. Other hyperparameters matched the standard Flow Matching configuration.

