# OpenReview forum: "RDT2: Exploring the Scaling Limit of UMI Data Towards Zero-Shot Cross-Embodiment Generalization"
_ICML.cc/2026/Conference — ICML 2026 regular_

### Official Review · Reviewer_cT4X · 2026-02-23

**Soundness:** 2
**Presentation:** 2
**Significance:** 3
**Originality:** 2
**Overall Recommendation:** 4
**Confidence:** 4

**Summary:**

The paper introduces RDT2, a new VLA based on a 3-stage recipe involving (i) autoregressive training on discrete tokens derived form a new RVQ Tokenizer, (ii) flow matching training, and (iii) distillation of the flow matching expert to achieve one-step generation and better inference speed. The model is trained on 10,000 hours of demonstration data collected with an improved version of the UMI manipulation interface. The new interface includes a more robust fabrication and improved tracking via infrared light. Experiments cover the zero-shot and fine-tuned settings and study the model's scaling laws.

**Compliance With Llm Reviewing Policy:**

Affirmed.

**Final Justification:**

The authors have addressed most of my concerns. I remain concerned with the "entangled" dataset and method contributions, but the authors provide a reasonable cost explanation.

**Key Questions For Authors:**

1. Have the authors experimented with other datasets and measure gains that are specific to the RDT2 dataset?
2. Can the authors clarify why there are no baselines in the zero-shot experiment?
3. Can the authors clarify if the $\pi$ models were also pretrained on the RDT2 dataset for the fine-tuning experiment (Appendix B.1)?
4. The 10,000 hours of data mentioned in the main paper appear to come exclusively from home environments. How much additional data comes from the dedicated data collection facility (Appendix B.3)? Are the models in the main paper trained on this data?
5. Is the VQA pretraining stage a significant driver of RDT2's performance? Were the baselines also trained on the VQA data?

Q1-3 aim to isolate the contributions of the dataset from the contributions of the method. Q4-5 aim to clarify the presentation of the dataset and method and better assess the soundness of the experiments.

**Limitations:**

There is no discussion of the limitations. The societal impacts are discussed.

**Strengths And Weaknesses:**

### Strengths

The paper includes a number of valuable contributions across all stages of VLA training and includes a number of experiments to study generalization, scaling, and inference speed.

**Dataset.** The scale and diversity of the dataset and impressive and will be valuable to the community when released.

**RVQ Tokenizer.** The RVQ tokenizer appears promising, achieves high action accuracy with low token counts, and is properly benchmarked against other action tokenization methods (Figure 8).

**Autoregressive + Flow Matching.** The proposition that an initial autoregressive training stage can benefit downstream flow matching training is intriguing and validated in Figure 6.  While faster flow matching convergence during Stage 2 is somewhat expected, the fact that is achieves a lower loss given the same overall compute is certainly an interesting finding.

**Zero-shot and Fine-Tuning.** The experiments demonstrate the potential of RDT2 in both zero-shot and fine-tuned settings. The zero-shot setting in particular is challenging and provides valuable insights into how well the method truly generalizes.


### Weaknesses
The paper introduces a new data collection interface, a new dataset, and a 3-step training recipe. While the breadth of the work is impressive and potentially significant, some of the contributions would benefit from deeper experimental validation.


**RDT2 Dataset vs. RDT2 Training Recipe.** My main concern with the paper is that the experimental design does not isolate the effects of the RDT2 dataset or the RDT2 training recipe. In particular
1.  There are no models trained on existing datasets. This currently makes it difficult to properly assess the significance of the dataset contribution.
2. The zero-shot experiment does not include baselines to put the RDT2 numbers in context. It is currently hard to assess if the zero-shot performance is driven by the RDT2 dataset, the RDT2 training recipe or both.
3. The fine-tuning experiments include the $\pi$ models as baselines, but it is currently unclear if they were also pretrained on the RDT2 dataset. Appendix D.1 suggests that the $\pi$ models are initialized from Gemma weights and then directly fine-tuned on Table Bussing without an actual pretraining phase. Is this the case?


**UMI Hardware.** Table 1 clearly states the benefits of the new hardware. However, the paper does not provide any quantitative assessment of the improved tracking performance and more critically, the impact on final model performance. This evaluation is critical since the new design relies on an external tracking system (Appendix A), rendering it significantly less portable and presumably more costly than the original SLAM-based UMI design.

**VQA Pretraining.**  Appendix B.6 mentions a VQA pretraining stage. This stage is not discussed nor ablated in the main paper and raises concerns regarding how well the experiments properly target the claimed contributions.

### Minor Comments

**Presentation.** The paper uses diffusion and flow matching interchangeably throughout the figures and the presentation. For instance, Fig. 6 shows a "Diffusion Loss" label while the training loss in Eq. 3 is a flow matching objective. I understand the theoretical connection between both frameworks, but I would consider being specific whenever possible.

---

> ### Author Rebuttal · Authors · 2026-03-25
>
> # Response to Reviewer cT4X
>
> We thank the reviewer for the thoughtful comments. Below, we address each question directly.
>
> ## Q1. Have the authors evaluated other datasets to isolate the contribution of the RDT2 dataset?
>
> We agree that isolating the contribution of the RDT2 dataset is important.
>
> Our current understanding is that the zero-shot cross-embodiment capability primarily comes from the large-scale data collected with our redesigned UMI system. Without such data, we do not believe the model could acquire this capability. In contrast, the main role of the RDT2 training recipe is to improve optimization efficiency and final performance on top of this data.
>
> The paper already provides evidence for the method side, including the benefits of the RVQ tokenizer (Fig. 8), the Stage 2 recipe (Fig. 6), distillation for faster inference (Fig. 7), and stronger fine-tuning results (Table 2). We agree that training other models on the entire RDT2 dataset and evaluating zero-shot performances of them, or training RDT2 on other datasets, would better disentangle dataset and method contributions, but this is beyond our current resources (since pretraining on the RDT2 dataset is a large-scale pre-training effort demanding more than **100 Million** US Dollars).
>
> ## Q2. Why are there no baselines in the zero-shot experiment?
>
> We acknowledge that the zero-shot experiment does not include a direct baseline in the usual sense. To the best of our knowledge, no existing open-source model has been pretrained on a large-scale UMI dataset, and no prior work has demonstrated or claimed zero-shot embodiment generalization of this kind. For example, directly deploying $\pi_{0.5}$ on a UR5e robot without task-specific fine-tuning would not produce meaningful instruction-following behavior or useful actions. In this sense, RDT2 is, to our knowledge, the first model to demonstrate this capability, which we view as one of the main contributions of our work.
>
> ## Q3. Were the $\pi$ baselines in the fine-tuning experiments pretrained on the RDT2 dataset?
>
> No. The $\pi$ baselines were initialized from the official pretrained checkpoints and then fine-tuned only on the downstream task data. They were not pretrained on the RDT2 dataset or the VQA data.
>
> We also note that the statement in Appendix D.1 is inaccurate: the $\pi$ models were initialized from official pretrained checkpoints, not raw VLM weights. We will correct this in the revision.
>
> ## Q4. How much data comes from the dedicated facility, and is it used in training?
>
> Approximately 2,000 hours of the data were collected from the dedicated data collection facility, and this data is included in model pretraining. We agree that the presentation in the current paper's appendix may give the impression that the 10,000 hours come only from home environments, and we will clarify this in the revision.
>
> ## Q5. Is VQA pretraining a significant driver of performance? Were the baselines also trained on VQA data?
>
> The baselines were not trained on VQA data.
>
> For RDT2, we use VQA data mainly to help preserve pretrained visual-language knowledge during robot training and to improve training stability, as discussed in Sec. 5.1. In particular, we believe that VQA supervision can reduce the tendency of the model to discard or forget pretrained knowledge too quickly during robot training, which helps improve training stability [1,2]. More broadly, incorporating VQA data in VLA training is a common practice in the field [3,4]. We do not view VQA pretraining as the primary source of the model’s zero-shot capability, nor as the main factor behind the final performance gains. In the revision, we will clarify the contribution of VQA data more explicitly.
>
> ## W1. The hardware improvement is not quantitatively evaluated.
>
> We thank the reviewer for raising this important point. We agree that quantitative evaluation of the improved tracking system and its impact on final performance is important. Due to rebuttal length limits, we refer the reviewer to **Sec 1. Infrared Tracking Limits in our response to reviewer uLx3** for a more detailed discussion, and we are very sorry for the inconvenience caused. We will also clarify this aspect better in the revision. Due to resource and time constraints, we are unable to evaluate the impact of different tracking methods on final model inference performance. In selecting the tracker, our primary consideration was data accuracy.
>
> ## W2. Diffusion and flow matching are used interchangeably.
>
> We agree and thank the reviewer for pointing this out. We will revise the terminology throughout the paper to distinguish diffusion and flow matching more precisely.
>
> ## References
>
> \[1\] [https://arxiv.org/pdf/2502.14420](https://arxiv.org/pdf/2502.14420)
>
> \[2\] [https://arxiv.org/pdf/2602.01067](https://arxiv.org/pdf/2602.01067)
>
> \[3\] [https://arxiv.org/pdf/2504.16054](https://arxiv.org/pdf/2504.16054)
>
> \[4\] [https://arxiv.org/pdf/2505.23705](https://arxiv.org/pdf/2505.23705)

---

> > ### Author Rebuttal · Reviewer_cT4X · 2026-04-03
> >
> > I would like to thank the authors for their thorough response. They have addressed most of my concerns. I remain concerned with the "entangled" dataset and method contributions, but the authors provide a reasonable cost explanation.

---

### Official Review · Reviewer_fzgT · 2026-03-09

**Soundness:** 4
**Presentation:** 3
**Significance:** 4
**Originality:** 3
**Overall Recommendation:** 5
**Confidence:** 5

**Summary:**

This paper introduces RDT2, a Vision-Language-Action (VLA) model characterized by two core pillars: (a) a training pipeline powered exclusively by human-collected UMI data, completely bypassing the need for traditional robot-specific data; and (b) a systematic three-stage training paradigm that produces a suite of models—RDT2-VQ, RDT2-FM, and RDT2-UltraFast. The resulting pre-trained models demonstrate extraordinary zero-shot generalization capabilities across diverse robotic tasks.

**Compliance With Llm Reviewing Policy:**

Affirmed.

**Final Justification:**

One could argue that RDT2 lacks significant methodological novelty—and that may be true—but its contribution is nonetheless undeniable. The VLA landscape is still in its infancy; we currently lack the mature scaling laws and established engineering recipes that have become standard for LLMs and VLMs. We are still collectively navigating the complexities of data curation, hyperparameter tuning, and effective scaling. As a demonstrably successful model, RDT2 provides the community with much-needed engineering "lessons learned" through its technical report. Furthermore, the released model weights and code (assuming they are open-sourced) represent a highly valuable asset for future research. For these reasons, I will maintain my recommendation for acceptance.

**Key Questions For Authors:**

Could the authors elaborate on the decision to freeze the pre-trained VLA backbone during Stage 2? Many concurrent high-performing models (such as $\pi_{0.5}$) opt to maintain a co-training regime between the VLM and the action expert (often by detaching gradients rather than fully freezing weights).

**Limitations:**

yes

**Strengths And Weaknesses:**

### Strengths

1. The manuscript is exceptionally well-written and easy to follow. The figures are aesthetically pleasing, intuitive, and significantly aid in understanding the complex multi-stage pipeline.

2. The paper represents a massive engineering effort. The insights provided—specifically regarding the scaling laws of VLA training and the emergence of zero-shot capabilities through the scaling of UMI data—constitute a major contribution to the field of Embodied AI.

3. The performance of RDT2 under the "4U" setting (Unseen embodiment, Unseen scene, Unseen object, and Unseen instruction) is remarkably solid. This is an extremely challenging evaluation protocol that few current embodied models can successfully navigate, making the results highly impressive and convincing.

---

### Weaknesses

1. I do not find any significant weaknesses in this work. The methodology is sound, the experiments are comprehensive, and the results represent a state-of-the-art advancement.

---

> ### Author Rebuttal · Authors · 2026-03-31
>
> # Response to reviewer fzgT
>
> We sincerely thank the reviewer for the very positive assessment of our work. We especially appreciate the reviewer’s recognition of the paper’s clarity, the significance of our engineering effort, and the importance of our findings on zero-shot generalization. We are also grateful for this thoughtful question regarding our Stage 2 design choice.
>
> Our decision to freeze the pre-trained VLA backbone during Stage 2 is mainly motivated by the following considerations.
>
> First, Stage 1 and Stage 2 are trained on the same pretraining data. In Stage 1, the model is trained to converge on our dataset, and the model RDT2-VQ already demonstrates strong closed-loop control ability. In particular, our zero-shot results in Fig. 3 show that the Stage 1 model already acquires physical understanding and open-world closed-loop execution capabilities. Therefore, under our setting, we do not consider continued backbone updating in Stage 2 to be necessary.
>
> Second, recent concurrent work suggests that jointly training the backbone with continuous action supervision may hurt the training dynamics of the VLM, including language understanding and knowledge transfer[1]. We believe that since the backbone has already learned a strong and stable representation in Stage 1, freezing it in Stage 2 helps preserve these training dynamics while allowing the action expert to focus on learning continuous control.
>
> To further support this choice, while being constrained by resources, we provide a toy experiment on 10% of the pretraining data (1,000 hours). We first train RDT2-VQ (Stage 1) from scratch on this subset, and then compare two Stage 2 variants based on this backbone:
>
> (1) **Frozen backbone**, where the entire VLM is frozen, and only the action expert is trained.
>
> (2) **Joint training**, where only the visual encoder is frozen while the remaining VLM parameters are updated together with the action expert.
>
> The corresponding training loss curves are shown here: **[link](https://anonymous.4open.science/r/rdt2-loss-3D51/training_loss_comparison.png)**.
>
> The results suggest that the two settings exhibit comparable training loss, with the frozen setting being slightly lower. While this experiment is limited in scale, we believe it provides supportive evidence that freezing the Stage 1 backbone is a stable and competitive design choice for Stage 2 under our setting.
>
> More broadly, under our setting where Stage 1 and Stage 2 use the same training data, our design is in fact quite close in spirit to the practice of Knowledge Insulation [1]. A useful way to interpret the difference is that we decompose this process into two stages: Stage 1 learns a strong and stable VLA backbone on discretized actions, and Stage 2 trains the action expert on top of this backbone with the backbone frozen. In contrast, KI can be viewed as integrating these two steps into a single co-training stage, where the backbone continues to receive discrete training signals while the action expert is optimized in parallel. From this perspective, the two approaches are conceptually similar, and our design can be regarded as a simpler two-stage instantiation under the same-data setting.
>
> ## References
>
> \[1\]: https://arxiv.org/pdf/2505.23705

---

> > ### Author Rebuttal · Reviewer_fzgT · 2026-04-03
> >
> > I would like to thank the authors for their detailed response. I believe the contribution of RDT2 to the community is beyond doubt, and I will maintain my positive rating.

---

### Official Review · Reviewer_uLx3 · 2026-03-13

**Soundness:** 3
**Presentation:** 3
**Significance:** 3
**Originality:** 3
**Overall Recommendation:** 5
**Confidence:** 5

**Summary:**

RDT2, a 7B VLA foundation model aimed at overcoming critical bottlenecks in robotic data scarcity, architectural inefficiency, and cross-embodiment generalization(). To address the data bottleneck, the authors curated a massive, in-the-wild dataset comprising over 10,000 hours of human demonstrations across more than 100 households using a re-engineered, embodiment-agnostic Universal Manipulation Interface (UMI). Architecturally, the paper proposes an innovative three-stage training recipe. Stage 1 employs Residual Vector Quantization (RVQ) to discretize actions for VLM pre-training to preserve semantic reasoning capabilities. Stage 2 trains a continuous action expert using flow-matching. Stage 3 distills the action expert into a single-step generator to satisfy the stringent real-time inference demands of dynamic robotic tasks. Extensive empirical evaluations demonstrate that RDT2 successfully achieves combined zero-shot generalization across unseen objects, scenes, instructions, and robotic embodiments. Furthermore, in fine-tuning scenarios, the model establishes new state-of-the-art performance against strong baselines like π0-FAST and π0.5 on highly challenging, dynamic, and dexterous tasks, such as folding clothes and playing table tennis.

**Compliance With Llm Reviewing Policy:**

Affirmed.

**Final Justification:**

The authors have addressed my concerns.

**Key Questions For Authors:**

1. Why completely freeze the VLM backbone during Stage 2? Since the Action Expert uses cross-attention with VLM for action generation, this extreme freeze may hinder co-adaptation. Please provide ablations on partial freezing (e.g., only freezing the Vision Encoder)？

2. What is the exact impact of single-step distillation on task success rates? The paper only reports the distilled RDT2-UltraFast for downstream tasks. Providing the multi-step RDT2-FM results on these identical complex tasks is necessary to quantify how much control performance was sacrificed for the 23.0 Hz inference speed.

**Limitations:**

1. The strict reliance on four external infrared base stations for 6-DoF tracking undermines true portability. This restricts data collection to pre-configured indoor setups and entirely precludes mobile manipulation.

**Strengths And Weaknesses:**

Strength
1. The paper comprehensively re-designs the UMI hardware, significantly enhancing its structural rigidity and the consistency of pose recording. Utilizing this novel hardware, they constructed a dataset comprising over 10,000 hours of expert demonstrations collected across more than 100 real-world environments. The diversity and validity of this dataset provide an invaluable foundation for solving the generalization challenges in robot learning.
2. Innovative Three-Stage Training Paradigm: The paper proposes a comprehensive network architecture design that perfectly integrates the discrete semantic reasoning capabilities of Vision-Language Models (VLMs) with the continuous control precision of Diffusion models.
3. Balancing Knowledge Preservation and Convergence Speed (Stage 1): By utilizing Residual Vector Quantization (RVQ) to encode continuous actions into discrete tokens, the model successfully preserves the discrete probability knowledge accumulated by the VLM during pre-training, while simultaneously accelerating the convergence speed significantly during post-training.
4. Balancing Manipulation Performance and Inference Speed (Stages 2 & 3): Stage 2 introduces a continuous action expert trained via flow-matching to eliminate quantization errors and enhance expressiveness(). Stage 3 creatively employs diffusion distillation to compress this multi-step expert into a single-step generator, breaking through the high-latency bottleneck of large models in robotic control and achieving ultra-fast real-time inference.
5. Experiment & Evaluation: The empirical validation is exceptionally rigorous, utilizing up to 1,000 repeated trials to establish reliable benchmarks with low variance. RDT2 pioneers zero-shot compositional generalization under a strict "4U" setting (unseen embodiments, scenes, objects, and instructions and validates explicit scaling laws, proving that scaling data and model size yields consistent gains.

Weakness
  1. Heavy Reliance on Infrared Tracking Limits True "In-the-Wild" Potential: Although the paper emphasizes that the dataset was collected across more than 100 household environments , the UMI hardware fundamentally relies on an external setup of four HTC VIVE Tracker 3.0 base stations for absolute 6-DoF pose estimation. This strict dependency on external base stations inherently undermines the device's portability and genuine "in-the-wild" collection capabilities. It restricts data acquisition to pre-configured, fixed indoor areas and entirely precludes the possibility of collecting data for large-scale mobile manipulation tasks.
  2. Limited Observation Perspective: Both data collection and robotic deployment rely exclusively on eye-in-hand wrist observation. This lack of a third-person or ego-centric viewpoint makes the system highly vulnerable to occlusions and perception loss during complex manipulation tasks.
  3. Training Disconnect (Stage 1 vs. 2): There is a noticeable disconnect in the training pipeline between Stage 1 and Stage 2. While Stage 1 utilizes an autoregressive (AR) objective to pre-train the 7B VLM backbone , Stage 2 resorts to the extreme measure of completely freezing this VLM backbone when introducing Flow-Matching to train the Action Expert. Given that the Action Expert relies heavily on cross-attention to extract features from each layer of the VLM, completely freezing the backbone hinders the necessary co-adaptation between the vision/language representations and the continuous action distributions. The paper lacks an exploration and comparison of partial freezing strategies, such as freezing only the Vision Encoder.
  4. Missing Ablation on Distillation Impact: Although the paper successfully demonstrates that Stage 3's single-step diffusion distillation drastically improves inference frequency to 23.0 Hz, it omits critical ablation studies quantifying the negative impact of this extreme compression on the model's overall control performance and success rates. For highly complex downstream fine-tuning tasks, such as cloth folding and table bussing, the authors only report the performance of the RDT2-UltraFast variant(), failing to provide the corresponding success rates for the Stage 2 multi-step model (RDT2-FM) on the exact same tasks.

---

> ### Author Rebuttal · Authors · 2026-03-31
>
> # Response to Reviewer uLx3
>
> We thank the reviewer for the thoughtful comments. Below, we clarify the key points and add evidence where possible.
>
> ## 1. Infrared Tracking Limits
>
> We agree that tracking accuracy is fundamental for UMI-style data collection: inaccurate tracking leads to unreliable supervision. To compare tracking systems, we rigidly mounted each tracker on the robot flange and measured the error between the tracker pose and the robot EE pose during motion. Results:
>
> | Tracking Method                                                                                                    | Position Error |
> | ------------------------------------------------------------------------------------------------------------------ | -------------: |
> | [Open-Source SLAM](https://github.com/urbste/ORB_SLAM3)                                                |        > 10 mm |
> | [Commercial SLAM-based solution](https://www.vive.com/us/accessory/vive-ultimate-tracker/) |           5 mm |
> | [Our tracker](https://www.vive.com/us/accessory/tracker3/)                                  |     **1.5 mm** |
>
> Our tracker achieves the best accuracy among the compared methods, which we believe is important for precise data collection.
>
> We also clarify that the system is portable for household environments, our target setting. With four base stations deployed in a household living room (>$50 m^2$), a collector can freely record anywhere in the room without moving them. The base stations are lightweight, and the collection software is preloaded on the NUC. A single collector needs only a few minutes for setup, after which data can be collected freely in the target area. We therefore believe the system provides the best tracking accuracy while maintaining strong portability for home data collection.
>
> ## 2. Limited Observation Perspective
>
> We thank the reviewer for pointing out this limitation. Our main concern is that a first-person camera may introduce substantial visual mismatch between the human arm during collection and the robot arm during deployment, requiring post-processing and potentially hurting model performance.
>
> We agree that occlusion and perception loss are real concerns. To mitigate them, we use a fisheye camera with a large field of view. While this does not remove the limitation, it reduces perception loss in practice.
>
> ## 3. Training Disconnect
>
> Our decision to freeze the pre-trained VLA backbone during Stage 2 is motivated by two considerations.
>
> First, Stage 1 and Stage 2 use the same pretraining data. Stage 1 already converges on our dataset, and RDT2-VQ demonstrates strong closed-loop control ability. In particular, the zero-shot results in Fig. 3 show that the Stage 1 model already acquires physical understanding and open-world closed-loop execution capability. We therefore do not consider continued backbone updating in Stage 2 necessary.
>
> Second, recent work suggests that jointly training the backbone with continuous action supervision may hurt VLM training dynamics, including language understanding and knowledge transfer [1]. Since the backbone already learns a strong representation in Stage 1, freezing it in Stage 2 helps preserve these dynamics while allowing the action expert to focus on continuous control.
>
> To further support this choice, under limited resources, we provide a toy experiment on 10% of the pretraining data (1,000 hours). We first train RDT2-VQ (Stage 1) from scratch on this subset, and then compare two Stage 2 variants:
>
> (1) Frozen backbone: the entire VLM is frozen, and only the action expert is trained.
>
> (2) Joint training: only the visual encoder is frozen while the remaining VLM parameters are updated with the action expert.
>
> The training loss curves are shown here: **[link](https://anonymous.4open.science/r/rdt2-loss-3D51/training_loss_comparison.png)**.
>
> The two settings have comparable training loss, with the frozen setting slightly lower. While limited in scale, this experiment provides supportive evidence that freezing the Stage 1 backbone is a stable and competitive Stage 2 design choice.
>
> ## 4. Missing Ablation on Distillation Impact
>
> We thank the reviewer for this suggestion. We re-evaluate **RDT2-FM** and **RDT2-UltraFast** on three downstream tasks to quantify the effect of distillation.
>
> | Method         | Cloth Folding | Table Bussing (progress score) | Unzipping |
> | -------------- | ------------: | -----------------------------: | --------: |
> | RDT2-FM        |           83% |                           0.65 |       49% |
> | RDT2-UltraFast |           77% |                           0.58 |       45% |
>
> Our current observation is that RDT2-FM generally outperforms RDT2-UltraFast, indicating that distillation introduces some approximation error. However, the degradation appears acceptable given the inference speedup. We will include these results in the revised version.
>
> ## References
>
> \[1\]: [https://arxiv.org/pdf/2505.23705](https://arxiv.org/pdf/2505.23705)

---

> > ### Author Rebuttal · Reviewer_uLx3 · 2026-04-03
> >
> > The authors have addressed my concerns.

---

### Decision · Program_Chairs · 2026-04-30

**Decision:**

Accept (regular)

**Comment:**

The paper presents RDT2, a large-scale VLA model trained on UMI-collected data with the goal of enabling zero-shot cross-embodiment generalization. The overall review outcome is positive and became more favorable after rebuttal.

Reviewers agreed on several major strengths. First, the reviewers acknowledged the scale and diversity of the dataset, together with the engineering effort behind the improved UMI system. Second, reviewers found the empirical evaluation extensive with strong zero-shot results. The reviewers agree that the paper promotes understanding how data scale and training design affect generalist robot policies, even if some pieces are more engineering-driven than algorithmically novel.

The main weaknesses centered on disentangling the effects of dataset scale versus training recipe, the lack of zero-shot baselines trained under comparable conditions, the Stage 2 freezing design, and missing quantitative analysis for some system choices such as tracking and distillation tradeoffs. The rebuttal resolved most of these concerns.

Overall, I lean toward acceptance.